# Retrograde ERK activation waves drive base-to-apex multicellular flow in murine cochlear duct morphogenesis

**Mamoru Ishii[1], Tomoko Tateya[2], Michiyuki Matsuda[1,3], Tsuyoshi Hirashima[1,4,5]***

[1]Graduate School of Biostudies, Kyoto University, Kyoto, Japan; [2]Department of Speech and Hearing Sciences and Disorders, Faculty of Health and Medical Sciences, Kyoto University of Advanced Science, Kyoto, Japan; [3]Graduate School of Medicine, Kyoto University, Kyoto, Japan; [4]The Hakubi Center, Kyoto University, Kyoto, Japan; [5]Japan Science and Technology Agency, PRESTO, Kawaguchi, Japan

**Abstract** A notable example of spiral architecture in organs is the mammalian cochlear duct, where the morphology is critical for hearing function. Genetic studies have revealed necessary signaling molecules, but it remains unclear how cellular dynamics generate elongating, bending, and coiling of the cochlear duct. Here, we show that extracellular signal-regulated kinase (ERK) activation waves control collective cell migration during the murine cochlear duct development using deep tissue live-cell imaging, Förster resonance energy transfer (FRET)-based quantitation, and mathematical modeling. Long-term FRET imaging reveals that helical ERK activation propagates from the apex duct tip concomitant with the reverse multicellular flow on the lateral side of the developing cochlear duct, resulting in advection-based duct elongation. Moreover, model simulations, together with experiments, explain that the oscillatory wave trains of ERK activity and the cell flow are generated by mechanochemical feedback. Our findings propose a regulatory mechanism to coordinate the multicellular behaviors underlying the duct elongation during development.

*For correspondence:
hirashima.tsuyoshi.2m@kyoto-u.ac.jp

**Competing interests:** The authors declare that no competing interests exist.

## Introduction

Spiral shapes are a widely occurring motif in many varied biological tissues and organisms, including shells, horns, and plants, but it has remained unclear how spiral shapes are formed spontaneously (*Thompson, 1942*). The general principle of spiral formation is differences in growth rate between the outer and inner tissues of the extending organ, with the growth rate of the outer tissue being faster than that of the inner one, which has been theoretically and experimentally demonstrated in shells and plants (*Johnson et al., 2019*; *Raup and Michelson, 1965*; *Smyth, 2016*; *Wada and Matsumoto, 2018*). The cellular processes causing this differential tissue growth are unique to each organ (*Johnson et al., 2019*; *Saffer et al., 2017*), and so identifying the organ-specific mechanisms underlying the differential tissue growth is crucial to understanding the developmental process of spiral morphogenesis.

An example of a spiral organ is the mammalian cochlear duct, which is a tonotopically organized auditory organ in the inner ear (*Figure 1A*). During murine development, the cochlear duct, composed of epithelial cells, elongates, bends, and coils to form a spiral. The molecular basis for the morphogenesis of the cochlear duct has been the subject of several previous studies. Gene knockout studies have clarified that the elongation of the cochlear duct requires sonic hedgehog (SHH) signaling from the cochleovestibular ganglion in the conical central axis of the cochlea (*Bok et al., 2013*; *Liu et al., 2010*; *Tateya et al., 2013*), fibroblast growth factor (FGF) signaling of epithelial cells (*Pauley et al., 2003*; *Pirvola et al., 2000*; *Urness et al., 2018*; *Urness et al., 2015*), and non-

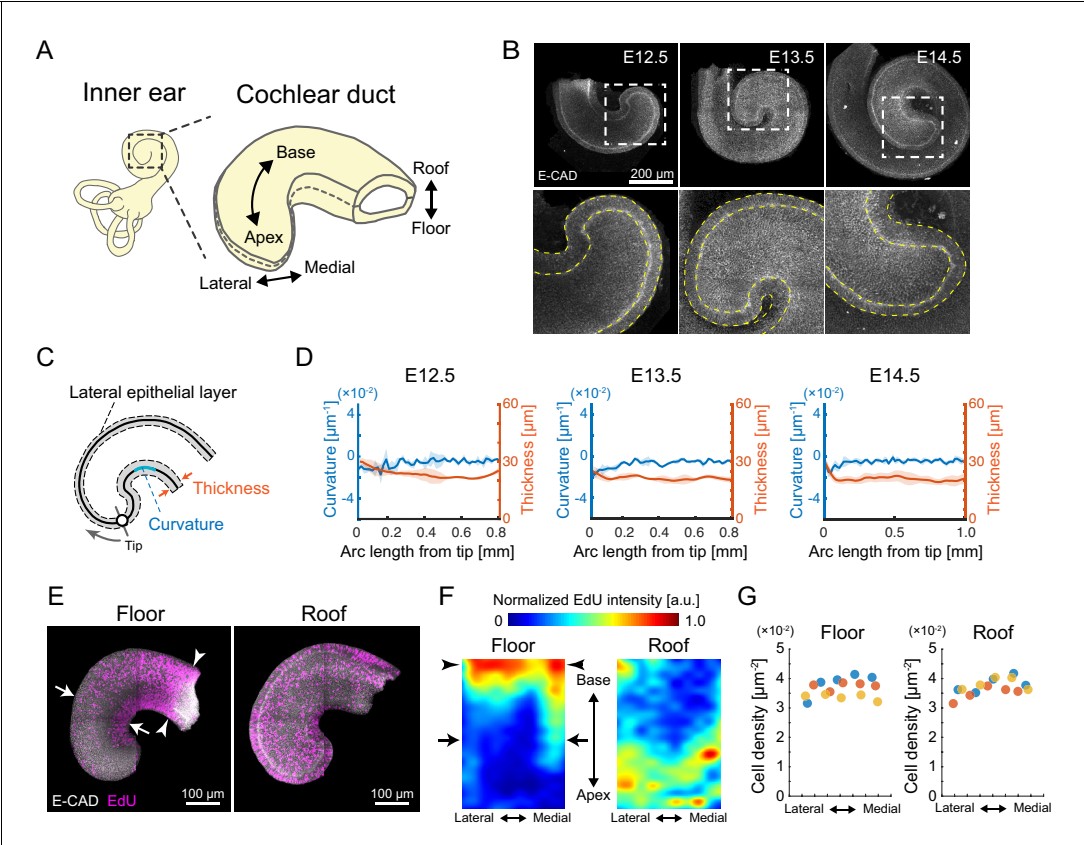

**Figure 1.** Quantification of morphology and cell proliferation in developing murine cochlear duct. (**A**) Schematic diagrams showing the tissue axis and labels of the cochlear duct. (**B**) Immunofluorescence images of anti-E-cadherin staining in the murine developing cochlea from E12.5 to E14.5. The lower rows are magnified images of the dotted squares in the upper rows. Yellow dotted lines represent the edges of the epithelial layer. Scale bar, 200 μm. (**C**) Schematic diagram showing regions used for morphological quantification. (**D**) Curvature and thickness as a function of the arc length from the apex tip along the lateral epithelial layer from E12.5 to E14.5. Mean ± standard deviation (s.d.) N = 3. (**E**) Maximum projection images of stained anti-E-cadherin (white) and EdU (magenta) in the roof and floor region of cochlear duct at E12.5. Representative images showing EdU signals in the cochlear duct are shown. Arrows indicate the region of EdU signal gradient from the medial to the lateral side of the duct. Arrowheads indicate the base region where the EdU intensity is concentrated. Scale bar, 100 μm. (**F**) Heatmap of the sample-mean of maximum EdU intensity projection in the roof and floor region of the duct. Arrows and arrowheads are the same as in (**E**). N = 3. Heatmaps of summed and mean intensity projection are shown in *Figure 1— figure supplement 1*. (**G**) Cell density distribution along the mediolateral axis in the floor and roof region, respectively. Color indicates the different samples. N = 3.

The online version of this article includes the following source data and figure supplement(s) for figure 1:

**Source data 1.** Curvature and thickness from E12.5 to E14.5.
**Source data 2.** EdU intensity profiles.
**Source data 3.** Cell density along the medial–lateral axis.
**Figure supplement 1.** Heatmaps of EdU (EdU: 5-ethynyl-2'-deoxyuridine, DMSO: dimethyl sulfoxide, PFA: paraformaldehyde, PBS: phosphate-buffered saline) intensity.

canonical Wnt–planar cell polarity (PCP) signaling of prosensory cells (*Mao et al., 2011*; *Montcouquiol and Kelley, 2020*; *Qian et al., 2007*; *Saburi et al., 2008*; *Wang et al., 2005*). Deletion of *Shh* expression leads to a shortening of the cochlear duct, and it was proposed that the SHH signaling promotes growth of the cochlear duct mainly in the base and middle regions (*Bok et al., 2013*). In *Fgf10* null mutant mice, the cochlear duct is remarkably shorter, but cell proliferation is unaffected (*Urness et al., 2015*), suggesting that cell proliferation and other cellular processes regulate ductal outgrowth. From embryonic day (E) 14.5 onward, the mediolateral active migration of prosensory cells, during which these cells intercalate radially with their neighbors (known as convergent extension), contributes to longitudinal duct extension in a Wnt–PCP pathway-dependent manner (*Chen et al., 2002*; *Cohen et al., 2020*; *Driver et al., 2017*; *Yamamoto et al., 2009*). This cell

intercalation drives ductal elongation; however, it simply cannot explain the duct bending before E14.5 without an asymmetric mode of cell intercalation (*Sato et al., 2015*). Although the underlying signaling pathways are well characterized, the physical cellular mechanisms underlying spiral morphogenesis of the cochlear duct remain elusive. In the present study, we aimed to identify the multicellular dynamics giving rise to elongating, bending, and coiling of the developing cochlear duct using a combination of live-cell imaging, Förster resonance energy transfer (FRET) quantitation, and mathematical modeling.

## Results

### Cell proliferation profile suggests cellular inflow to the lateral side of the apex for elongation of bending duct

We first examined the morphology of the developing cochlear duct from E12.5 to E14.5 by staining an epithelial marker, E-cadherin, followed by organ-scale 3D imaging. During this developmental period, the cochlear duct elongates and coils without changes to the mediolateral width at a horizontal section of the roof–floor axis (*Figure 1B*). The curvature and thickness of the epithelial layer was quantified on the lateral side of the cochlear duct (*Figure 1C*). We found that they remained almost constant and small variation along the arc length from the apex tip (*Figure 1D*, *Figure 1— source data 1*), suggesting a cellular mechanism for the developing duct to elongate while maintaining its curvature.

Next, we examined the spatial distribution of proliferating cells in the duct as it directly contributes to the local tissue growth. The spatial distribution of nuclei labeled with EdU for 30 min was measured on the roof and floor side of the cochlear duct at E12.5 (*Figure 1E*). For quantification, the image domain of the cochlear duct was divided into interrogation regions and the averaged fluorescence intensity of labeled EdU was measured within each region. On the floor side, EdU-positive cells were more abundant in the medial side than in the lateral side around the apex (arrows, *Figure 1E, F*, left, *Figure 1—source data 2*). However, on the roof side of the cochlear duct, EdU-positive cells were distributed evenly, with slightly fewer cells at the base than at the apex and without a significant bias along the mediolateral axis (*Figure 1E, F*, right, *Figure 1—source data 2*). Moreover, nuclear density showed no significant difference along the mediolateral axis in either the floor or the roof side (*Figure 1G*, *Figure 1—source data 3*). These observations do not strongly support the possibility that cell proliferation rates on the lateral side of the cochlear duct could drive mediolateral differential tissue growth and cause duct bending. Of note, the spatial map of EdU intensity shows that cell proliferation rates were higher in the floor–base region (arrowheads, *Figure 1E, F*, left). Supposing that cell proliferation is the main driver of local tissue growth, the higher volumetric growth observed in the medial side than in the lateral side would contribute to the duct bending inward at the lateral side, which contradicts the innate cochlear morphogenesis. We thus hypothesized that the cells in the lateral side of the growing apex may be supplied by the proliferation 'hot spot' in the base region of the cochlear duct, which could resolve the observed mismatch between tissue growth rates in the medial and lateral sides of the duct.

### ERK inactivation resulted in cochlear duct shortening

We then investigated how the cells are supplied from the base to the apex of the cochlear duct. Earlier studies reported that FGF signaling is critical for cochlear duct outgrowth (*Pirvola et al., 2000*; *Urness et al., 2015*). Therefore, we focused on extracellular signal-regulated kinase (ERK)/MAP kinase, a downstream kinase in the FGF signaling pathway, and investigated the impact of FGFR–ERK signaling axis on the cochlear duct morphogenesis.

The dissected cochlea at E12.5 were cultured ex vivo for 2 days by treating with either PD0325901, a specific ERK kinase (MEK) inhibitor (*Barrett et al., 2008*), or SU5402, a multitargeted receptor tyrosine kinase inhibitor including FGFR (*Sun et al., 1999*). Both inhibitor treatments resulted in obvious impairments of the cochlear duct growth together with the reduction of EdU signals in the entire cochlear duct, indicating that the FGF–ERK signaling promotes the cell proliferation (*Figure 2A–A''*). The arc length of the cochlear duct on the lateral side from the apex tip to the adjacent point of the saccule was significantly shorter both in the PD0325901 and SU5402 treatments than in the control (*Figure 2B*, *Figure 2—source data 1*). These results suggest that the cell

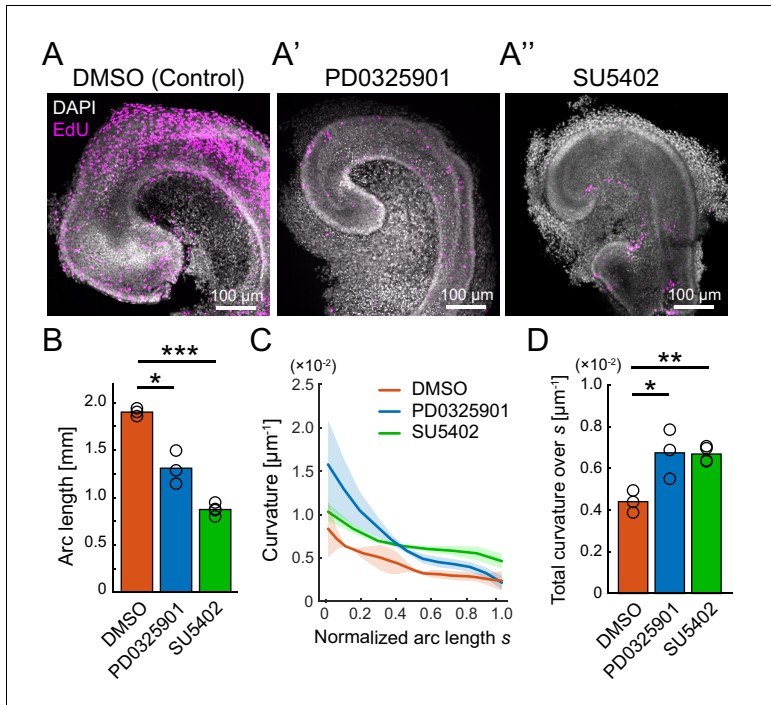

**Figure 2.** Pharmacological inhibition of FGFR–ERK signaling axis resulted in the impairment of cochlear duct growth. (A–A'') Maximum projection images of EdU labeling (magenta) with DAPI (white) nuclear counterstaining over the entire cochleae after 2 days of ex vivo culture from E12.5 in the treatment with DMSO (A), PD0325901 at 1 µM (A'), and SU5402 at 30 µM (A''). Scale bar, 100 µm. (B) The arc length of the cochlear duct in the treatment with DMSO (red), PD0325901 (blue), and SU5402 (green). N = 3. Two-sample t-test without assuming equal population variances. p=0.023 for DMSO-PD0325901 and p<0.001 for DMSO-SU5402. (C) The curvature of the cochlear duct along the normalized arc length s. Mean ± s.d. N = 3. (D) Total curvature over the normalized arc length in the treatment with DMSO (red), PD0325901 (blue), and SU5402 (green). Two-sample t-test without assuming equal population variances. p=0.021 for DMSO-PD0325901 and p<0.0033 for DMSO-SU5402. ERK: extracellular signal-regulated kinase.

The online version of this article includes the following source data for figure 2:

**Source data 1.** Longitudinal length for treatments with DMSO, PD0325901, and SU5402.
**Source data 2.** Curvature over arc length from the tip for treatments with DMSO, PD0325901, and SU5402.

proliferation would contribute to the duct elongation. Moreover, we found that those inhibitor treatments remarkably altered the curvature of the cochlear duct. As the size of cochlear duct was different in each treatment, the curvature profile along the arc length was normalized by the total arc length (*Figure 2C*, *Figure 2—source data 2*). To assess the curvature over the lateral edge of cochlear duct, we calculated total curvature over the normalized arc length, s, and found that the total curvature was larger both in the PD0325901 and SU5402 treatments than in the control (*Figure 2D*). Provided that cell proliferation is the only driving factor for the cochlear duct bending, the curvature profile should be almost constant along the lateral edge as observed at E12.5 (*Figure 1D*). However, the inhibitor treatments for the FGFR–ERK signaling resulted in the clear curvature gradient with larger curvature in the apex region than in the base region (*Figure 2C*). Therefore, the growth impairment of the cochlear duct by inhibition of FGFR–ERK signals would be caused not only due to the cell proliferation but also other cellular processes, and we suspected that the FGF–ERK signaling would play a role in the cell supply from the base to the growing apex region in the cochlear duct.

# Retrograde helical ERK activation waves drive base-to-apex multicellular flow

To examine the spatiotemporal ERK activity on the developing cochlea, we used a reporter mouse line that ubiquitously expresses a FRET-based biosensor for ERK activity in the cytosol (*Harvey et al., 2008*; *Komatsu et al., 2018*; *Komatsu et al., 2011*). 3D FRET imaging using two-photon microscopy revealed that ERK is preferentially activated in the lateral-roof side of the cochlear duct, including the outer sulcus and stria vascularis (*Figure 3A, A'*, *Figure 3—video 1*), which is consistent with the previously reported distribution of *Fgfr2* expression (*Urness et al., 2015*).

For continuous observation during ductal outgrowth, we established an explant culture method in which the capsule above the apex tip was partially removed, allowing 3D organ-scale long-term imaging of ERK activity. Surprisingly, the time-lapse images of the cochlea dissected at E12.5 revealed that ERK activation propagates intercellularly as oscillatory waves from the apex to the base of the floor side (*Figure 3B*, *Figure 3—video 2*), while ERK is constitutively activated around the apex tip of the roof side (*Figure 3—figure supplement 1A, B*). We next quantified multicellular tissue flow by particle image velocimetry (PIV) at the supracellular (4–5 cell length) scale and found that cells coherently move as clusters of ~100 μm diameter from the base to the apex of the floor side, again as oscillatory waves, and similarly to the ERK activation waves of the floor side (*Figure 3C*, *Figure 3—video 2*). In the roof side, the multicellular tissue flows directly toward the direction of elongation around the apex tip (*Figure 3—figure supplement 1C*). Kymography of ERK activity along the apex–base line of the lateral-floor side (*Figure 3D*) shows oscillatory retrograde ERK activity waves (*Figure 3E*, *Figure 3—figure supplement 1D*), which proceed at a speed of $0.42 \pm 0.078$ μm min$^{-1}$ (mean ± standard deviation) in space-fixed coordinates (*Figure 3F*).

Since ERK activation can be induced by cell extension – an increase in the projected area along the cellular apicobasal axis – during collective cell migration (*Hino et al., 2020*), we then calculated the ERK activity rate (time derivative of ERK activity) and the extension-shrinkage rate, that is, the local tissue strain rate along the apex–base line, by using time-series data of ERK activity and PIV speed, respectively (*Figure 3—figure supplement 1E*). We found that both the ERK activity rate and the extension-shrinkage rate oscillate across the time course (*Figure 3G*, *Figure 3—source data 1*). Moreover, cross-correlation analysis revealed that the local tissue deformation, represented by the extension-shrinkage rate, precedes the ERK activity rate by 24 min on average (*Figure 3H*).

The role played by ERK was confirmed via inhibitor assays at E12.5. Treatment with either PD0325901 or SU5402 resulted in the significant decrease of the tissue flow speed, as well as ERK inactivation (*Figure 3—figure supplement 1F, F'*), that is, the median speed decreased before and after the administration by 42% and 48%, respectively (*Figure 3I*, *Figure 3—source data 2*). These results corroborated that the FGFR–ERK signaling axis contributed to the base-to-apex multicellular tissue flow. On the other hand, we found that cyclopamine treatment, an inhibitor for the Shh signaling pathway, resulted in almost no change in the ERK activity (*Figure 3—figure supplement 1F, F'*, *Figure 3—video 3*) and affected the cell flow speed (16% reduction) to a lesser extent compared with PD0325901 and SU5402 treatment within 3 hr after administration (*Figure 3J*, *Figure 3—source data 2*). This suggested that ERK activity was not the only factor but it played a major role in regulating cell migration. Interestingly, we found that the cell flow was decreased by 62% along with ERK inactivation 8 hr after the administration (*Figure 3J*, *Figure 3—source data 2*, *Figure 3—video 3*). This long-term effect both on the cell flow and ERK activity might be caused by the lack of cell supply from the base to the apex due to the suppression of the Shh-induced cell proliferation.

Finally, we extended the analysis to 3D dynamics of ERK activity and cell movement in the developing cochlear duct. Surface rendering of the ERK activity map in the cytosol indicated that ERK activity peaks shift from the apex–roof to the base–floor in the lateral side of the cochlear duct (*Figure 3K*, *Figure 3—video 4*). Concomitantly with helical ERK activity waves, coherent cell movements can be observed from the base–floor to the apex–roof in the opposite direction to the ERK waves (*Figure 3—video 4*). This observation suggests that ERK-mediated helical collective cell movement on the lateral side could drive 3D duct coiling underlying the spiral morphogenesis of the cochlear duct (*Figure 3L*).

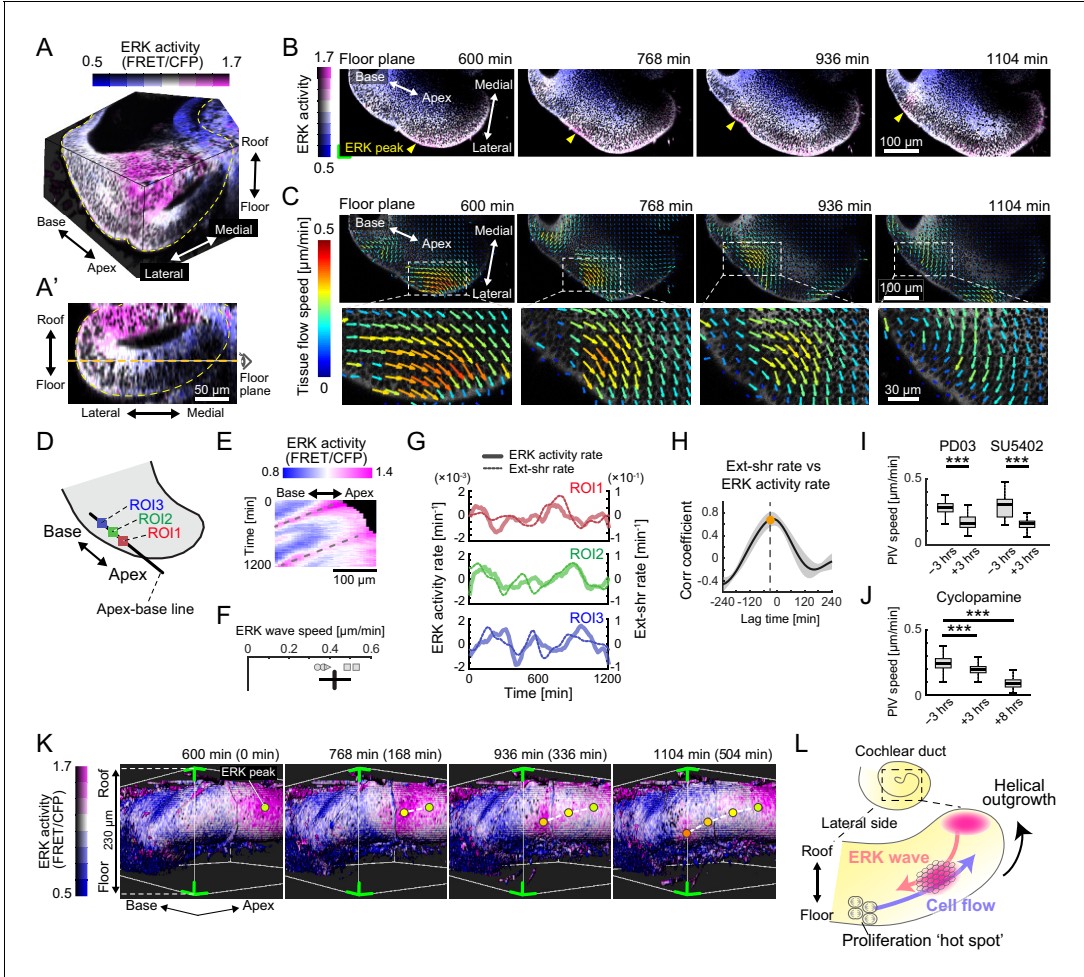

**Figure 3.** Retrograde helical extracellular signal-regulated kinase (ERK) activation waves drive base-to-apex multicellular flow. (**A**) 3D ERK activity map in the cochlear duct cultured ex vivo for 1 day from E12.5. (**A′**) Cross-sectional view (medial–lateral and roof–floor plane) of (**A**). Orange dotted line indicates the floor plane shown in (**B**) and (**C**). Scale bar, 50 μm. (**B**) Time-lapse snapshots of ERK activity maps in the floor plane. Time indicates the elapsed time of live imaging. Yellow arrowheads indicate the ERK activity peak. Scale bar, 100 μm. (**C**) Time-lapse snapshots of tissue flow speed obtained by particle image velocimetry in the floor plane. Scale bar, 100 μm. (**D**) Schematic diagram showing the axis, the apex–base line for kymography, and regions of interests (ROIs). (**E**) Representative kymograph of ERK activity. The horizontal axis indicates the position on the apex–base line shown in (**D**), and the vertical axis indicates the elapsed time of live imaging. Dotted lines represent oscillatory waves from the apex to the base. Scale bar, 100 μm. (**F**) ERK wave speed with mean and s.d. n = 5 from N = 3. (**G**) Time-series ERK activity rate and extension-shrinkage rate in representative three different ROIs. (**H**) Cross-correlation between the extension-shrinkage rate and ERK activity rate. n = 12. Mean ± s.d. (**I, J**) Tissue flow speed before and after the PD0325901 treatment at 1 μM, the SU5402 treatment at 30 μM (**I**), and the cyclopamine treatment at 30 μM (**J**). n = 285. Confirmed by N = 2. Two-sample t-test, p<0.001. (**K**) Time-lapse snapshots of surface-rendered ERK activity maps in the cochlear duct at E12.5. The green corners correspond to the green corner on the images shown in (**B**) and viewed from the left-bottom corner of (**B**). Circles indicate the position of ERK activity peaks, and the connecting dotted lines indicate a trace of the peak shift. The timescale is the same as in (**B**). (**L**) Schematics for the ERK activity waves and cell flow.

The online version of this article includes the following video, source data, and figure supplement(s) for figure 3:

**Source data 1.** Extracellular signal-regulated kinase activity rate and extension-shrinkage rate.

**Source data 2.** Particle image velocimetry speed before and after the treatments with PD0325901, SU5402, and cyclopamine.

**Figure supplement 1.** Extracellular signal-regulated kinase (ERK) activity waves and cell flows.

**Figure 3—video 1.** 3D extracellular signal-regulated kinase (ERK) activity map of the cochlear at E12.5, related to *Figure 3*.
https://elifesciences.org/articles/61092#fig3video1

**Figure 3—video 2.** Time-lapse movie of the extracellular signal-regulated kinase (ERK) activity and multicellular tissue flow, related to *Figure 3*.
https://elifesciences.org/articles/61092#fig3video2

**Figure 3—video 3.** Time-lapse movie of extracellular signal-regulated kinase (ERK) activity upon treatment with cyclopamine, related to *Figure 3*.
https://elifesciences.org/articles/61092#fig3video3

*Figure 3 continued on next page*

eLife Research article

Developmental Biology

## ERK-mediated mechanochemical feedback explains cell flow and ERK waves

The cross-correlation analysis and ERK inactivation assay suggested a regulatory regime of coupling between the ERK activation and cell migration; the extension-triggered ERK activation promotes cell contraction and pulling the neighboring cells, which eventually evokes transmission of ERK activation to the neighboring cells, that is, mechanochemical feedback (*Boocock et al., 2021*; *Hino et al., 2020*). To further verify this regime underlying the developing cochlear duct, we first suppressed actomyosin cell contraction by treating the E12.5 cochlea with blebbistatin. As anticipated, the cochlear duct was extended immediately after blebbistatin treatment and ERK was activated (*Figure 4A*, *Figure 4—video 1*). Moreover, the cell flow was significantly decreased by 79% (*Figure 4B*, *Figure 4—source data 1*), indicating that the active cellular contraction was required for the cell flow.

Next, we noticed local deformation of the cochlear duct as a possible consequence of cellular contraction. Our live imaging system from E12.5 exhibited that the epithelium of the cochlear duct became concave locally in the high ERK activity region (*Figure 4C*), suggesting that the ERK activation promoted contraction. We then calculated the tissue curvature of the ductal lateral edge using time-lapse images and found that both the ERK activity and the tissue curvature similarly oscillated over the time (*Figure 4D*, *Figure 4—source data 2*). Cross-correlation analysis revealed that the curvature change was delayed by 24 min on average due to the changing ERK activity (*Figure 4E*).

Finally, we confirmed the plausibility of the mechanochemical coupling via mathematical modeling (*Figure 4F*). Our minimal mathematical model of the mechanochemical coupling reproduced multiple ERK activity propagations well (*Figure 4G*) as observed experimentally (*Figure 3E*). The mechanochemical model produced a 28 min lag from the extension-shrinkage rate to the ERK activity rate (*Figure 4H, I*). In a counterpart uncoupling model (*Figure 4J*), however, it was a 2 min lag under which the ERK activation waves regulated cell deformation unidirectionally (*Figure 4K–M*). Together, the model analysis supported the plausibility of mechanochemical coupling rather than uncoupling regulation.

## Discussion

Previous genetic studies have revealed the molecular basis of cochlear duct elongation during development, but have been unable to explain the physical mechanisms by which the duct bends because of its severe phenotype (*Bok et al., 2013*; *Groves and Fekete, 2012*; *Urness et al., 2018*; *Urness et al., 2015*). Motivated by these earlier studies, we have visualized the cochlear duct development by two-photon microscopy under ex vivo culture condition. We found that the coherent multicellular flow occurs from the base to the apex exclusively on the lateral side of the growing cochlear duct and also elucidated that the multicellular flow was accompanied by retrograde ERK activation waves (*Figure 3L*). Thus, our long-term deep tissue imaging has illuminated unprecedented dynamics of cells and kinase activity underpinning the bending of developing cochlear duct.

In the present study, the establishment of long-term imaging techniques and biosensors for protein kinase activity has led to the discovery of unexpected spatiotemporal patterns of cell movement and ERK activity in the developing cochlear duct. Previously, we and others have observed intercellular ERK activation waves in the epithelium, such as migrating Madin–Darby canine kidney (MDCK) cells (*Aoki et al., 2017*; *Hino et al., 2020*), developing *Drosophila* tracheal placode (*Ogura et al., 2018*), and wounded murine skin (*Hiratsuka et al., 2015*). We have also proposed an ERK-mediated mechanochemical feedback system, in which cell extension activates ERK followed by ERK-triggered cell contraction (*Boocock et al., 2021*; *Hino et al., 2020*) that can explain the coordination between cell movement and ERK activity. The ERK activation wave speed in the developing cochlear duct was 0.42 μm min$^{-1}$ (*Figure 3F*), which is slower than in MDCK cells and wounded mouse epidermis,

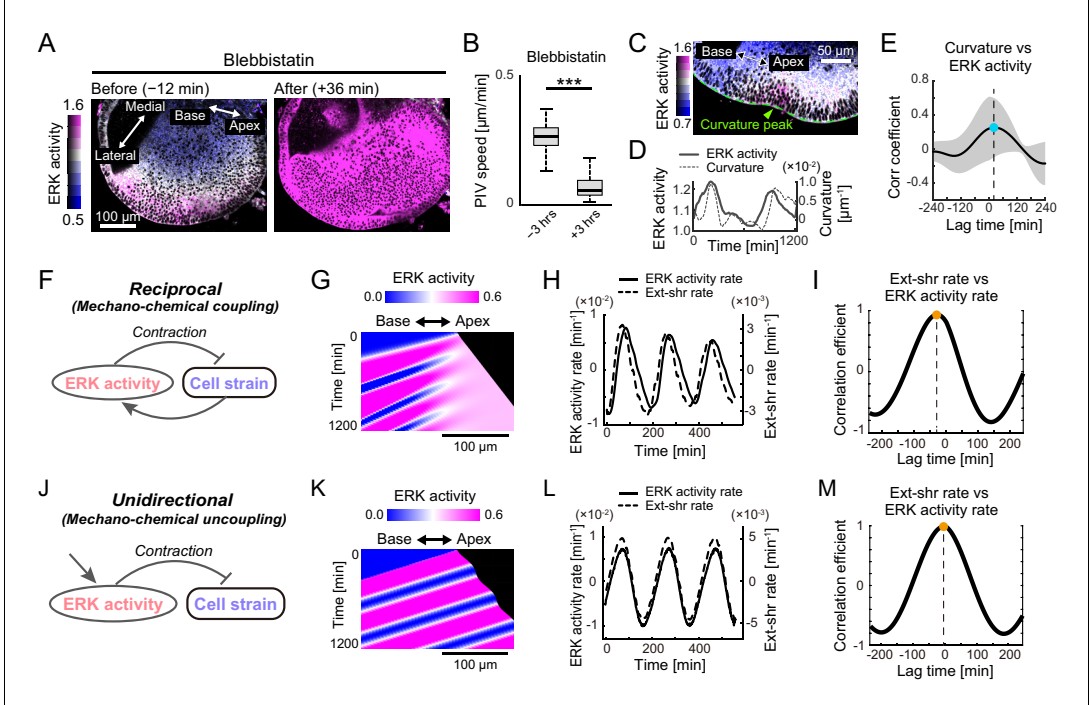

**Figure 4.** Extracellular signal-regulated kinase (ERK)-mediated mechanochemical feedback explains cell flow and ERK waves. (A) ERK activity maps in the floor plane before (−12 min) and after (+36 min) the treatment with 30 µM of blebbistatin. Time indicates the timing of blebbistatin treatment. Scale bar, 100 µm. (B) Tissue flow speed before (−3 hr) and after (+3 hr) blebbistatin treatment. N = 258. Two-sample t-test, p<0.001. (C) Snapshot of the time-lapsed ERK activity map on the lateral side of the cochlear duct (green dotted line), showing coexistence of the ERK activity peak and the tissue curvature peak (green arrowhead). Scale bar, 50 µm. (D) Time-series ERK activity and the tissue curvature in the representative regions of interest. (E) Cross-correlation between the tissue curvature and ERK activity. n = 9. Mean ± s.d. (F) Schematics for the model mechanochemical coupling, in which ERK activity and cell deformation are reciprocally regulated. (G) Kymograph of the ERK activity in the model simulation. Scale bar, 100 µm. (H) Simulated time series of ERK activity rate and extension-shrinkage rate in the mechanochemical coupling regime. (I) Cross-correlation between the extension-shrinkage rate and the ERK activity rate. The lag time is −28 min. (J) Schematics for a counterpart regime of the mechanochemical coupling, in which the ERK activity unidirectionally regulates the cell deformation without closed feedback. (K) Kymograph of the ERK activity in the uncoupling model simulation. Scale bar, 100 µm. (L) Simulated time series of ERK activity rate and extension-shrinkage rate in the uncoupling regime. (D) Cross-correlation between the extension-shrinkage rate and the ERK activity rate. The lag time is −2 min.

The online version of this article includes the following video, source data, and figure supplement(s) for figure 4:

**Source data 1.** Particle image velocimetry speed before and after the treatments with blebbistatin.

**Source data 2.** Extracellular signal-regulated kinase activity and tissue curvature.

**Figure supplement 1.** Multicellular tracking of the time-lapse images.

**Figure 4—video 1.** Time-lapse movie of extracellular signal-regulated kinase (ERK) activity upon treatment with blebbistatin, related to *Figure 4*.
https://elifesciences.org/articles/61092#fig4video1

where it proceeds at 2.5 µm min$^{-1}$ and 1.4 µm min$^{-1}$, respectively (*Aoki et al., 2017*; *Hino et al., 2020*; *Hiratsuka et al., 2015*). Interestingly, when normalized to the cell lengths of the developing cochlear duct (4 µm), MDCK cells (20 µm), and the basal cells of the mouse epidermis (10 µm), the wave speed of the developing murine cochlear duct, 6 cells diameter h$^{-1}$, is comparable with that of MDCK cells, 7 cells diameter h$^{-1}$, and that of wounded adult murine skin, 8 cells diameter h$^{-1}$. Hence, our findings on the coupling between the multicellular flows and the ERK activation wave trains in the cochlear duct, together with the earlier studies (*Boocock et al., 2021*; *Hino et al., 2020*), support the existence of a general regulatory mechanism for the collective cell migration during tissue morphogenesis.

Recently, much slower intercellular ERK activation wave propagation was found in the regeneration of zebrafish scales (*De Simone et al., 2021*); the ERK activation wave speed is about at 0.17 µm min$^{-1}$, that is, 1 cell diameter h$^{-1}$ with the average osteoblast size (10 µm). It is proposed that these slow ERK waves were generated by a reaction–diffusion system including

diffusible ligand-based activators and inhibitory regulators, such as dual-specificity phosphatases and sprout proteins. Of note, the reactions in this system consist of the ERK-mediated transcriptional and translational processes that characterize the slower timescale of the dynamics in the cells and the ERK activity compared with the other reported phenomena. Moreover, the diffusion of an Fgf ligand as hypothesized for scale regeneration in zebrafish was not considered in the present study since the mechanochemical cell-to-cell communications, which transmit the signals across the cells via diffusion of mechanical stress, can sufficiently explain the ERK activation waves for the cochlear duct development. As the ligand diffusion-based signal transmission is directly affected by 3D tissue geometry, especially in the curved epithelial tissues, the mechanical signal transmission would achieve a more robust regulatory system underlying the morphogenesis of the spiral cochlear duct.

Our 3D time-lapse imaging revealed coherent helical cell flow from the base–floor to the apex–roof in the lateral side of the cochlear duct. Cell flow analysis revealed that the rate of base-to-apex cell flow (0.24 μm min$^{-1}$, *Figure 3—figure supplement 1E*) exceeds that of the duct elongation speed (0.13 μm min$^{-1}$, *Figure 3—video 2*). Thus, the cell flow rate may be sufficient to compensate for the lateral tissue growth. We speculate that this ERK-mediated cell advection originating from the heterogeneity of cell proliferation causes consistent mediolateral differential growth at the tissue scale and results in cochlear bending. In support of this, knockout of *Shh* causes a significant decrease in the number of proliferating cells at the base of the cochlear duct and shortens the cochlear duct (*Bok et al., 2013*). Moreover, we showed that the impact of cyclopamine treatment on the cell flow became less evident immediately after its administration (*Figure 3J*), although the slight reduction of cell speed implied that the Shh signaling could be involved in controlling machineries for cell migration. As SHH is secreted mainly from the spiral ganglions located in the central axis of the cochlea (*Bok et al., 2013*), further investigations on the interaction between the cochlear duct and these ganglions will be helpful to clarify the overall cell flow in the elongating duct.

We reported that the cell flow depended on myosin activity (*Figure 4B*) and caused the local deformation of the basal edge (*Figure 4C*). Therefore, the cells generated the mechanical forces and actively migrated to coordinate the multicellular behavior at supracellular scale in the lateral side of the cochlear duct. The epithelium in the lateral side is simple cuboidal rather than pseudostratified observed in the floor and medial side, thereby the epithelial cells would attach to the basement membrane and load forces onto the basement membrane via the subcellular structures, such as cryptic lamellipodia and cellular protrusions as observed in the floor side (*Driver et al., 2017*). However, it was unfeasible to identify the single-cell behavior due to mainly our fluorescence labeling for cytosol in this study. We also attempted to detect the F-actin cortex on the inner face of the cell membrane using Lifeact-EGFP transgenic mice (*Riedl et al., 2010*), but we were not able to recognize the cell membrane dynamics clearly enough owing to its faint signals in the deep region. This problem will be solved by further improvements in fluorescence markers, spatiotemporal high-resolution microscopy, and organ culture methods.

Although we focused on the cell flow on the lateral side, there are other cellular events that can potentially underlie the duct morphogenesis. For example, the cartilaginous capsule may play an essential role in cochlear morphogenesis, especially as a physical restriction to avoid outward duct growth. Without volumetric growth of the capsule, the cochlear duct would not be able to grow due to a lack of appropriate space. Moreover, complete removal of the capsule led to failure in the elongating and bending of the cochlear duct cultured under the ex vivo condition in our observations. Therefore, the detailed balance of overall tissue growth between the duct epithelium and the capsule will be essential and needs to be elucidated. Another potential but less likely mechanism is the active cell intercalation that occurs from E14.5 onward as demonstrated previously (*Chen et al., 2002*; *Cohen et al., 2020*; *Driver et al., 2017*; *Yamamoto et al., 2009*). Depending on our careful observation and manual cell tracing of time-lapsed imaging data from E12.5 to E14.5, there was no clear mediolateral intercalation (*Figure 4—figure supplement 1*). However, the mediolateral cell intercalation might contribute to the maintenance of the cochlear duct width through polarized cellular mechanoresponse as proposed in an elongating epithelial duct (*Hirashima and Adachi, 2019*). As our live imaging data were insufficient for automatic 3D cell tracking due to a lack of organelle-specific marker and weak fluorescence signals, further improvement of imaging systems and single-cell tracking will clarify these aspects of multicellular complexity.

Overall, we visualized multicellular behavior underlying the elongating and bending of the cochlear duct during development using deep tissue live imaging. This contributes to a better

understanding of symmetry breaking in tissue morphogenesis during development and in the generation of inner ear organoids (*Koehler et al., 2017*; *Koehler et al., 2013*). The live imaging technique used in the present study forms the basis for further analysis of the interplay between morphogenesis and cell fate decisions during cochlear development (*Cohen et al., 2020*; *Tateya et al., 2019*).

## Materials and methods

### Experiments

#### Animals
For FRET imaging, we used the transgenic mice that ubiquitously express an ERK biosensor with a long flexible linker (hyBRET-ERK-NLS) reported elsewhere (*Harvey et al., 2008*; *Komatsu et al., 2018*; *Komatsu et al., 2011*). Otherwise, we used ICR mice purchased from Japan SLC, Inc. We designated the midnight preceding the plug as embryonic day 0.0 (E0.0), and all mice were sacrificed by cervical dislocation to minimize suffering. All the animal experiments were approved by the local ethical committee for animal experimentation (MedKyo 19090 and 20081) and were performed in compliance with the guide for the care and use of laboratory animals at Kyoto University.

#### Antibodies
The following primary and secondary antibodies were used for immunofluorescence: anti-E-cadherin rat antibody (Thermo Fisher Scientific, #13-1900, 1:100 dilution) and Alexa Fluor 546-conjugated goat anti-rat IgG (H+L) antibody (Thermo Fisher Scientific, #A11081, 1:1000 dilution).

#### Small-molecule inhibitors
The following chemicals were used: blebbistatin (FUJIFILM Wako Pure Chemical Corporation, #021-17041), cyclopamine (FUJIFILM Wako Pure Chemical Corporation, #038-19311), SU5402 (FUJIFILM Wako Pure Chemical Corporation, #197-16731), and PD0325901 (FUJIFILM Wako Pure Chemical Corporation, #162-25291).

#### Whole-tissue staining and imaging
The cochleae were gently freed from the capsule, and the staining and clearing were performed according to an earlier study (*Hirashima and Adachi, 2015*). Briefly, the samples were fixed with 4% PFA in PBS overnight at 4°C and then blocked by incubation in 10% normal goat serum (Abcam, #ab156046) diluted in 0.1% Triton X-100/PBS (PBT) for 3 hr at 37°C. The samples were treated with primary antibodies overnight at 4°C, washed in 0.1% PBT, and subsequently treated with secondary antibodies conjugated to either Alexa Fluor 546 or Alexa Fluor 647 overnight at 4°C. For counter staining of nucleus, we used DAPI (Dojindo Molecular Technologies, #D523-10, 1:200 dilution). The samples were mounted with 10 μL of 1% agarose gel onto a glass-based dish (Greiner Bio-One, #627871) for stable imaging. Then, the samples were immersed with the CUBIC-R+ (Tokyo Chemical Industry Co., #T3741) solution for optical clearing. Images were acquired using the confocal laser scanning platform Leica TCS SP8 equipped with the hybrid detector Leica HyD with the ×40 objective lens (NA = 1.3, WD = 240 μm, HC PL APO CS2, Leica) and the ×20 objective lens (NA = 0.75, WD = 680 μm, HC PL APO CS2, Leica) and the Olympus FluoView FV1000 with the ×30 objective lens (NA = 1.05, WD = 0.8 mm, UPLSAPO30XS, Olympus).

#### EdU assay
For EdU incorporation to embryos, 200 μL of 5 mg/mL EdU in PBS was intraperitoneally injected to pregnant mice 30 min prior to dissection. For the incorporation to dissected cochleae, 10 μM of EdU was treated into the samples 1 hr prior to the chemical fixation. Before EdU detection, whole-tissue immunofluorescence of E-cadherin and counter nuclei staining with DAPI were performed. EdU was detected using the Click-iT EdU Imaging Kits (Thermo Fisher Scientific, #C10340). The samples were optically cleared with CUBIC-R+, and images were acquired by confocal microscopy as described above.

## Explant cultures

We cultured the dissected cochleae without removing the capsule unless otherwise noted. The cochleae were mounted on a 35-mm glass-based dish (Iwaki, #3910-035) with 1 μL of growth factor reduced Matrigel (Corning, #356231), and filled with 2 mL of a culture medium including FluoroBrite DMEM Media (Thermo Fisher Scientific, #A1896701) with 1% GlutaMAX (Thermo Fisher Scientific, #35050061) and 1% N2 Supplement with Transferrin (Holo) (FUJIFILM Wako Pure Chemical Corporation, #141-08941). The samples were incubated at 37°C under 5% $CO_2$.

## Live imaging for explants

For long-term organ-scale imaging, we partially cut off the capsule adjacent to the apex tip of cochlear duct using tweezers carefully and the semicircular canals were removed. The isolated cochlea was put onto the dish as described above. For microscopy, we used an incubator-integrated multiphoton fluorescence microscope system (LCV-MPE, Olympus) with a × 25 water-immersion lens (NA = 1.05, WD = 2 mm, XLPLN25XWMP2, Olympus). The excitation wavelengths were set to 840 nm for CFP (InSight DeepSee, Spectra-Physics). Imaging conditions for the FRET biosensor were as follows: scan size: 800 × 800 pixels; scan speed: 10 μs/pixel; IR cut filter: RDM690 (Olympus); dichroic mirrors: DM505 and DM570 (Olympus); and emission filters: BA460-500 for CFP and BA520-560 for FRET detection (Olympus).

## **Quantification and analysis**

### FRET image analysis

The median filter of 3 × 3 window was processed to remove shot noises, and the background signal was subtracted each in the FRET channel and the CFP channel. Then, the ratio of the FRET intensity to the CFP intensity was calculated by a custom-made MATLAB (MathWorks) script. In the scale bar, color represents the FRET/CFP ratio and brightness represents the fluorescence intensity of the FRET channel.

### Measurement of layer curvature and thickness

For 2D measurement of curvature and thickness, we first performed whole-mount immunofluorescence of E-cadherin to visualize the cochlear epithelium and acquired z-stack images by confocal microscopy as described above. Next, we manually traced the apical and basal sides of epithelial cells on the middle horizontal section of the roof–floor axis. The extracted epithelial layer was named as the lateral epithelial layer according to the side based on a manually chosen apex tip point. Then, the curve of the epithelial layer was obtained by the iterative skeletonization, and discrete points $(x_i, y_i)$ were sampled along the curves at regular intervals of 15 μm. Finally, fitting the discrete points with a cubic spline function, the function $S_i$ at an interval $[x_i, x_{i+1}]$ is denoted as

$$S_i(x) = a_i(x - x_i)^3 + b_i(x - x_i)^2 + c_i(x - x_i) + d_i$$

Due to a definition of curvature $\kappa(x) = S''(1 + S'^2)^{-3/2}$, the curvature from the spline function was calculated as

$$\kappa_i(x) = \frac{6a(x - x_i) + 2b}{\left(1 + \left\{3a(x - x_i)^2 + 2b(x - x_i) + c\right\}^2\right)^{3/2}}.$$

The curve of the convex/concave to the duct lumen was assigned as positive/negative in $\kappa$. We defined the layer thickness as a linear length connecting to the luminal and basal edge, which is vertical to the curve of the epithelial layer at sampling points.

### EdU intensity mapping

First, we separated 8-bit staining image stacks for E-cadherin and EdU into two regions, roof and floor, based on z position at the middle point, and performed three different projection methods onto the xy plane, including (1) maximum intensity projection, (2) summed intensity projection, and (3) mean intensity projection averaged within the cochlear duct. The mean intensity projection evaluates the EdU signals averaged only within the cochlear duct epithelium so that the denominators

can be changed depending on the epithelial thickness on each measurement region. Therefore, there would be different results between the mean intensity projection and summed intensity projection. Next, we binarized immunostaining signals for E-cadherin using Otsu's method with morphological operations and detected periphery of the cochlear duct with the MATLAB function 'bwperim'. The medial curve was defined by connecting between the apex tip and the end of medial edge, both of which were given manually, along the duct periphery. Similarly, the lateral curve was defined by connecting between the apex tip and the end of lateral edge along the duct periphery. Then, we marked points to make 20 bins at a constant distance each along the medial curve and lateral curve. By connecting the marked point of the medial curve and that of the lateral curve indexed by an order from the apex tip and dividing the lines into 10, we partitioned the cochlear duct into small regions for measurement. Finally, we measured the averaged intensity of EdU signal within each region and normalized by 255. The EdU intensity distribution was worth being considered as a marker of local tissue growth at a fixed moment in time due to cell proliferation and homogeneous cell density.

### Cell density measurement

Five equally divided sections along the mediolateral axis were set on the floor or the roof of the cochlear duct at E12.5. In each section, the supracellular region including more than hundred cells that overlapped each other was manually chosen. Then, the number of cells and area were measured within each region. The center of mass of the region was regarded as the position of that region.

### Tissue flow and ERK activity

To calculate velocity fields of cells in cochleae, we performed PIV-based image processing using a free code MatPIV (a GNU public license software distributed by Prof. Kristian Sveen in University of Oslo) that was applied to time-lapse images of the CFP channel. Velocity fields at time $T$ were computed by displacement between $T$ and $T+\Delta t$. $\Delta t$ was set as the sampling rate, 12 min. The size of the interrogation window was set to 40 pixels, approximately 25 μm, corresponding to 4–5 cell diameter, and the window overlap was set to 50%. The obtained velocity data were then smoothened via median filtering to eliminate peaky noises. We then obtained the 'tissue flow speed for the elongation' from PIV velocity vector projected onto the apex–base line depicted in *Figure 3D* and calculated the spatial derivative of the tissue flow speed for the elongation between two adjacent interrogation windows on the apex–base line according to the definition of a diagonal component of the strain rate. This quantity was smoothened using the MATLAB function 'smooth' to eliminate high-frequency components and defined as the extension-shrinkage rate. As for the ERK activity, we set thresholds in CFP images using Otsu's method within each interrogation window to extract the cytoplasmic region and calculated the mean FRET/CFP ratio in the binarized region. The ERK activity rate was calculated as the time derivative of the ERK activity. Cross-correlation analysis was performed using the MATLAB function 'xcorr'.

### Tissue curvature and ERK activity

The interrogation squared windows, each of which had 20 pixels (approximately 13 μm) per side, were set along the manually traced lateral edge of the cochlear duct. For the calculation of tissue curvature, the reference point on the traced lateral edge was determined as the nearest from each interrogation window. Four points centered on the reference point were sampled along the lateral edge at regular intervals of 19 μm, and the curvature at the reference point was calculated as described above (see Measurement of layer curvature and thickness). For the ERK activity, the measurement was performed as described above (see Tissue flow and ERK activity).

## Statistical analysis

The number of cells or region of interests analyzed (n) and the number of biological replicates (N) are indicated in the figure legends. No particular statistical method was used to predetermine the sample size. A minimum of N = 3 independent experiments was performed based on previous studies in the field (*Cohen et al., 2020*; *Driver et al., 2017*; *Tateya et al., 2019*). No inclusion/exclusion criteria were used, and all analyzed samples were included in the analysis. No randomization was performed. Statistical tests, sample sizes, test statistics, and p-values are described in the main text.

p-Values of less than 0.05 were considered to be statistically significant in two-tailed tests and were classified into four categories: *p<0.05, **p<0.01, ***p<0.001, and n.s. (not significant, i.e., p≥0.05).

## Software

For digital image processing, we used MATLAB (MathWorks) and Image J (National Institute of Health). For graphics, we used MATLAB (MathWorks), Imaris (Bitplane), and ImageJ (National Institute of Health). For statistical analysis, we used MATLAB (MathWorks).

## Mathematical model

### Modeling oscillatory ERK activation waves and cell flows

We built a minimal 1D mechanochemical coupling model for the collective cell migration based on our previous studies (*Boocock et al., 2021*; *Hino et al., 2020*). Cells, each indexed as $j=1,...,N$, are represented as a chain of springs, whose junctions including the boundaries are labeled as $i=1,...,N+1$, with elastic constant $k$, and each cell generates contractile force at the rear side of the cell to move to the front with a force $F$. That is, the cell contractile force with $j=N$ ($F_{j=N}$) is regarded as the force at the junction $i=N$ ($F_{i=N}$). Because the epithelial cells adhere to neighboring cells and thus transmit the elastic force with viscous frictions $\eta_c$, the dynamics of cell collectives is represented as

$$\begin{aligned} \eta_c \dot{x}_i &= k(\varepsilon_j - \varepsilon_{j+1}) + F_{j=i}, \\ \varepsilon_j &= (x_{i+1} - x_i)/L - 1, \text{ for } i = 1...N \text{ and } j = 1...N \end{aligned} \tag{1}$$

where $\varepsilon$ is the cell strain and $L$ the typical cell length, that is, 5 µm. At the front edge of the cells, that is, $i=N+1$, a self-propelling force $F_{tip}$ is generated, reflecting an elongation of the apex tip. Since the cells respond to stretching as activating the ERK, a coupling between the cell kinematics and the ERK activity is formulated as

$$\eta_E \dot{E}_j = \tanh(\alpha \varepsilon_j) - E_j, \tag{2}$$

where $\alpha$ denotes the sensitivity parameter and $\eta_E$ the timescale of the dynamics. Then, the ERK activity is converted to the self-contractile force represented as dynamics of the

$$\eta_F \dot{F}_j = \lambda E_j - F_j, \tag{3}$$

where $\lambda$ denotes the controlling parameter of amplitude and $\eta_F$ the timescale.

As for the uncoupling regime, ERK activity was given as the following traveling waves instead of *Equation (2)*:

$$E(X) = \left( \sin\left( \frac{\pi X}{w} + vt \right) + 1 \right) \times 0.5 \tag{4}$$

where $w$ is the characteristic length of ERK activation and $v$ the ERK activation speed.

The parameter $w$ was set as 84 µm, that is, the wavelength of the ERK activity is 168 µm, from *Figure 3E*, and $v$ was set as 0.42 µm min$^{-1}$, from *Figure 3F*.

## Numerical simulation

The ordinary differential equations were numerically solved by the forward Euler method with time step 0.01 using MATLAB. The number of cells $N$ was set as 1000, and one boundary $i=1$ was fixed and the other $i=N+1$ was the moving boundary condition. The biologically plausible parameter set was determined as $\eta_c = 40$ (nN min µm$^{-1}$), $k = 20$ (nN), $F_{tip} = 6$ (nN), $\alpha = 3$, $\eta_E = 30$ (min), $\eta_F = 10$ (min), and $\lambda = 9$ (nN) according to the present study and a previous study (*Serra-Picamal et al., 2012*).

## Code availability

The MATLAB code is available at https://github.com/tsuyoshihirashima/2020_cochlearduct/blob/main/erk1d.m; *Ishii, 2021*; copy archived at swh:1:rev: e81398f7827e8a7f91171191224e269cea4685f4.

## Acknowledgements
This work was supported by the JSPS KAKENHI 17KT0107 and 19H00993, the JST PRESTO JPMJPR1949 and CREST JPMJCR1654, and the Medical Research Support Center of Kyoto University. We thank Akane Kusumi for technical assistance, and Edouard Hannezo, Naoya Hino, and Yoshiko Takahashi for fruitful discussion.

## Additional information

### Funding

| Funder | Grant reference number | Author |
| --- | --- | --- |
| Japan Society for the Promotion of Science | 17KT0107 | Tsuyoshi Hirashima |
| Japan Society for the Promotion of Science | 19H00993 | Michiyuki Matsuda Tsuyoshi Hirashima |
| Japan Science and Technology Agency | JPMJPR1949 | Tsuyoshi Hirashima |
| Japan Science and Technology Agency | JPMJCR1654 | Michiyuki Matsuda |

The funders had no role in study design, data collection and interpretation, or the decision to submit the work for publication.

### Author contributions

Mamoru Ishii, Resources, Data curation, Software, Formal analysis, Validation, Investigation, Methodology, Writing - original draft; Tomoko Tateya, Conceptualization, Methodology, Writing - review and editing; Michiyuki Matsuda, Resources, Supervision, Funding acquisition, Writing - review and editing; Tsuyoshi Hirashima, Conceptualization, Resources, Data curation, Software, Formal analysis, Supervision, Funding acquisition, Validation, Investigation, Visualization, Methodology, Writing - original draft, Project administration, Writing - review and editing

### Author ORCIDs

Tomoko Tateya (iD) http://orcid.org/0000-0002-0342-3688
Michiyuki Matsuda (iD) https://orcid.org/0000-0002-5876-9969
Tsuyoshi Hirashima (iD) https://orcid.org/0000-0001-7323-9627

### Ethics

Animal experimentation: All the animal experiments were approved by the local ethical committee for animal experimentation (MedKyo 19090 and 20081) and were performed in compliance with the guide for the care and use of laboratory animals at Kyoto University.

### Decision letter and Author response

Decision letter https://doi.org/10.7554/eLife.61092.sa1
Author response https://doi.org/10.7554/eLife.61092.sa2

## Additional files

### Supplementary files

• Transparent reporting form

### Data availability

All data generated or analyzed during this study are included in the manuscript and supporting files. Source data files are provided.

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
