## [Decision Letter]

**Acceptance summary:**

It is a huge challenge to study the morphogenetic events that give rise to the snail-shaped mammalian cochlea. Through live imaging of FRET-based mouse cochlear explants and mathematical modeling, this study demonstrated that a concomitant wave of an oscillatory wave of ERK activity and cell flow in the opposite direction contributes to the extension of the developing cochlear duct. This work provides a conceptual framework for considering factors that are required for cochlear morphogenesis and contributes to the general understanding of complex organ formation.

**Decision letter after peer review:**

Thank you for submitting your article "Interplay between medial nuclear stalling and lateral cellular flow underlies cochlear duct morphogenesis" for consideration by *eLife*. Your article has been reviewed by Marianne Bronner as the Senior Editor, a Reviewing Editor, and three reviewers. The following individuals involved in review of your submission have agreed to reveal their identity: Andres Collazo (Reviewer #2); David Sprinzak (Reviewer #3).

The reviewers have discussed the reviews with one another and the Reviewing Editor has drafted this decision to help you prepare a revised submission.

The mammalian cochlear duct is a spiral-shaped organ. This study investigated the mechanisms underlying the bending of the cochlear duct. Using two-photon live imaging and mathematical modeling, it was reported that the bending of the cochlear duct is caused by stalling of nuclei in the luminal side of the medial cochlear duct during interkinetic nuclear migration. Using FRET-based imaging, cochlear duct elongation is attributed to an oscillatory wave of ERK activity originating from the cochlear tip.

All three reviewers were impressed by the imaging results. Although the reviewers and editors find the concept and approach interesting, blocking cell proliferation may be too crude a method to address the authors' hypothesis and many questions were raised by the results of blocking cell division. Second, the relationship between cell proliferation and ERK-driven migration is also unclear. Please see comments from reviewer #2 and #3 for specifics. Third, what is the relationship between SHH-induced proliferation and ERK activation as suggested by the authors (see comments from reviewer #1)? Additionally, it is problematic to illustrate a difference in the bending force between medial and lateral cochlear duct that is presumably occurring at E12.5 and E14.5 with a cochlear dissection at E17.5. The tissue architecture is completely different between E12.5 and E17.5. The surgery basically removed a specific region of the cochlear duct, the stria vascularis, rather than medial versus lateral halves of the cochlear duct.

We hope the authors will be able to address these questions and comments and submit a revision. The full reviews are appended below for your information.

Reviewer #1:

This is a fascinating manuscript that explores for the first time the potential mechanisms underlying cochlear morphogenesis. The authors have used a combination of modeling, beautiful imaging and ERK-FRET reporter mice reporter mice to suggest at least two processes may be at play in cochlear shaping – differential interkinetic nuclear migration and a cellular flow that appears to correlate with ERK activation.

I have no major concerns with this lovely piece of work. The imaging and quantification is meticulous, and the observations made by the authors are novel and will of great interest to cell biologists interested in morphogenesis, no just aficionados of the inner ear.

The one suggestion I would make is for the authors to clarify the relationship between cell proliferation and ERK activation. When they reference the inner ear literature, they point out that FGF pathway mutants have deficient cochlear morphogenesis and proliferation, and they hypothesize that FGF-induced ERK activation may be responsible for their propagating waves. However, they also reference work suggesting that cellular extension during collective migration can also induce ERK activation and also suggest SHH-induced proliferation as another causative factor in promoting ERK activation through proliferation. I think the authors should try and clarify this – both in their explanation, but also by comparing the effects of the MEK inhibitor PD0325901 on ERK activity and tissue flow speed (Figure 4I and Figure 3—figure supplement 1F) with the effects of the FGFR inhibitor SU5402, and also Shh inhibitors like cyclopamine. If the effects they see are directly due to FGF signaling, one would expect a change in ERK activation and cell flow with the same kinetics as with PD0325901. However, if Shh-induced proliferation is responsible, the change in ERK activation would take much longer to achieve. I think these experiments should be possible to do in a relatively short period of time.

Reviewer #2:

The paper by Hirashima and colleagues shows some interesting cellular mechanisms they conclude drive the spiraling and outgrowth of the mammalian cochlea. The two cellular mechanisms they propose are supported by experiments and modeling. The spiraling ERK wave and the contrasting movement of lateral cells was very intriguing. However, the ERK wave and lateral cell movements seem disconnected from the bending forces discussed. Are the authors saying that the ERK mediated lateral cell movements are important for cochlear growth while the MEL is important for the bending? The two mechanisms they discuss seem insufficient to explain all of cochlear spiraling. Other cellular mechanisms such as cell proliferation and convergent extension are mentioned but their roles are not incorporated into their discussion. Are they not required? How do they complement their results?

1) While the authors talk about bending forces, the paper has no measurements of the forces generated by different tissues. I also feel there are other cellular mechanisms that are mentioned but never incorporated into their proposed explanation for duct coiling such as convergent extension and actomyosin based basal shrinkage. Proliferation is discussed quite a bit but seems to be dismissed as a force. In the introduction they mention how Shh mediated proliferation is required for duct elongation while Fgf10 null mutants have a shortened duct yet normal proliferation. So what is the role for proliferation? Maybe they can answer this in the context of their interesting observation that there is more proliferation in the roof than the floor which would be predicted to bend the cochlea along that axis. When combined with the medial lateral bending could these two forces result in the spiraling? It also seems like this differential proliferation between the floor and roof was in more than just the epithelium, correct? Could the cartilaginous capsule around the duct guide the bending as well? In their culture experiments, if too much of the capsule was removed then normal duct development was disrupted.

2) Their demonstration that the bending forces are in the medial half is interesting but the only tissue whose mechanism is studied is the MEL. Could convergent extension in other medial tissues such as the prosensory domain (which Wang et al., showed was occurring in this tissue) and surrounding mesenchyme be the main force generator for the bending of the medial half of the cochlear duct? Does the MEL cultured by itself bend? They say that cell intercalation can drive ductal elongation but not bending (Introduction) but can't convergent extension occur asymmetrically in the tissue? Such as by occurring in the overlying medial mesenchyme but not in the medial epithelium. It should be noted that the bending by the epithelium does not have to provide high forces as long as the force provided by other tissues are similar across the medial lateral axis, the bending in the epithelium could bias the mass of tissue to bend.

3) The mathematical modeling for the luminal bending is less convincing than the mathematical modeling for the ERK and Cell flow coupling. The simulated curves in Figure 2K are quite different from the Experimental measure in Figure 2M, especially for the Mitomycin C condition. I feel that the values plugged in for the Numerical simulation, the standard parameter set were not well justified. What happened to the simulations as these values changed? Was the parameter space for acceptable values broad? In contrast the parameters for the numerical simulation of the ERK activation waves and cell flows were well justified. The parameters chosen might explain the big differences between simulation and experimental in Figure 2.

4) For the cell tracking experiments in the lateral region the resolution was 4-5 cells. The resulting cell flow patterns were very interesting but why didn't the authors track single cells? Segmenting individual cells via cytoplasmic labeling is much trickier but the nuclei are identifiable and the Imaris software they used in the paper has a cell tracking feature for such labeling. I would think that individual cell movements might provide more insights. In subsection “Retrograde helical ERK activation waves drive base-to-apex multicellular flow” they say they can see cell contractions which I assume is for individual cells? How were cell contractions identified? Video 5 was excellent and very informative. Do the cell flows correlate at all with the proliferation seen with Edu staining?

Reviewer #3:

The manuscript by Ishii et al., focuses on understanding how cellular dynamics drive the spiral shape of the cochlear duct in mammals. The authors use live imaging of inner ear explants to follow dynamics of interkinetic nuclear migration (IKNM) and ERK activity (using ERK FRET sensor) to track some of the processes that give rise to tissue bending during spiral duct formation. On the imaging side, the manuscript presents a technical tour de force, showing remarkable two photon imaging capabilities that provide insights into the dynamics underlying cochlear extension. These experiments reveal several new observations: (1) Medial epithelial layer (MEL) tends to bend more than the lateral epithelial layer (LEL) despite being more proliferative. (2) That nuclei of cells in the curved region of the cochlea tend to stay in the luminal side, following cell division, rather than migrate back to the basal side. (3) The cells migrate towards the apical lateral roof. (4) That there are orchestrated ERK waves that correlate with cell migration. Based on these observations and on mathematical modeling, the manuscript has two main claims: (1) that nuclear stalling on the luminal side following cell division leads to increased curvature which gives rise cochlear duct bending, and (2) that multicellular flow mediated by ERK signaling waves pushes cells towards the growing apex, supplying the cells required for luminal expansion. While the observations in the manuscript are certainly interesting, I worry however, that some of the claims are not sufficiently substantiated, and also the connection between the two observations is rather weak. Here are the detailed concerns:

1) The authors argue that cell cycle arrest results in a decrease in the curvature of the cochlear duct, which supports the hypothesis that luminal nuclear stalling promotes MEL bending. This is fine, but luminal nuclear stalling can be a result and not a cause. Since in a bent region, the basal side is more packed, this density gradient can be the cause of nuclei stalling at the luminal side. The fact that the curvature decreased but not diminished after cell cycle arrest could suggest that nuclear stalling is not required for bending, but rather reinforces it.

2) Since the authors discuss both cell proliferation and nuclear stalling, and cell migration, as forces that can drive bending and coiling, it hard to interpret the results of the mitomycin C experiment. Could it be that the tissue is less curved because there are less cells to supply the elongation tissue rather than less nuclear stalling? The authors should consider inhibiting either cell migration or the cytoskeletal machinery required for IKNM to dissect these effects.

3) The authors present a mathematical model to demonstrate that nuclear stalling in the luminal side results in bending. To model nuclear motion they use a parameter, γ, which controls the degree of basalward movement after IKNM. Modeling in such way means that other than γ=1, the nucleus never fully returns to the basal side, but if I understand correctly this is not the case, as even if the nuclei that stall at the luminal side, eventually return to the basal side.

4) Furthermore, for luminal nuclear stalling, the authors tracked only the nuclei of dividing cells. This makes the data in Figure 2D' much clearer. However, in their model the authors show only these nuclei and not all nuclei. In addition, they show many crowded nuclei in the model, yet this is not observed in the images provided in the manuscript. Therefore, it seems the model does not represent the morphology of the tissue properly. The authors should model the process with non-dividing cells at the basal side.

5) In subsection “Spatial heterogeneity of cell proliferation suggests cellular inflow to the lateral side of the apex to realize cochlear bending” the authors claim that the higher volumetric growth measured at the MEL should cause an opposite curvature relative to the innate one. This is true if EdU intensity is proportional to volumetric growth, but cells in the MEL and LEL may not be the same size. For example, cells in the MEL could be smaller than cells in the LEL. The authors should therefore measure the nuclei number density and the volumetric cell density to clarify this. If the number density of the nuclei is indeed higher at the MEL, it may also explain the higher structural integrity of the MEL relative to the LEL demonstrated in Figure 1C.

6) The authors show the effect of ERK inhibition on tissue flow speed. This is a very important observation and raises several important questions. What is effect of ERK inhibition on curvature? On tissue length? On proliferation? These will provide a more complete understanding of the effect of RK inhibition.

7) The authors should also test the effect of mitomycin C on cell flow and ERK activity. As mentioned above, it is not clear whether the effect of mitomycin C is a result of less nuclear stalling or perhaps less cells that flow towards the apex.

8) In Figure 3 the authors analyze the EdU distribution over the cochlear duct. This analysis is done using the maximum intensity projection of the stack. It seems that a more accurate way to quantify would be to use the summed intensity image rather than the maximum intensity image. This may reveal additional details that were missed by throwing away all other layers except the one at maximum intensity.

9) In Figure 4 the colors used for the ERK activity analysis are very hard to see for color-blind people. It would be easier for this audience if the authors changed one of these colors to green/red/yellow.

[Editors' note: further revisions were suggested prior to acceptance, as described below.]

Thank you for submitting your article "Interplay between medial nuclear stalling and lateral cellular flow underlies cochlear duct morphogenesis" for consideration by *eLife*. Your article has been re-reviewed by Marianne Bronner as the Senior Editor, a Reviewing Editor, and three reviewers.

The reviewers and the Reviewing Editor have discussed the reviews with one another and the Reviewing Editor has drafted this decision to help you prepare a revised submission.

All three reviewers have judged that your manuscript is of interest and represents an advancement to the field. However, the manuscript cannot be accepted at its current form because of the issue whether IKNM described is causal or consequential to cochlear bending. This issue needs to be better resolved. Either the mathematical model is revised to accommodate for fewer cells stalling at the luminal side but could still account for the cochlear bending or more cells can be observed to stall at the luminal side that could account for the cochlear bending. Alternatively, the entire manuscript can be totally revised to accommodate for these two possibilities in an unbiased manner, starting with the Title.

Since many researchers have temporarily lost access to the labs, we will give authors as much time as they need to submit revised manuscripts. We are also offering, if you choose, to post the manuscript to bioRxiv (if it is not already there) along with this decision letter and a formal designation that the manuscript is "in revision at *eLife*". Please let us know if you would like to pursue this option. (If your work is more suitable for medRxiv, you will need to post the preprint yourself, as the mechanisms for us to do so are still in development.)

Specific comments from two of the reviewers are listed below as guidelines for your revision:*Reviewer #2:*

The authors have addressed most of my comments. The experiments with the MEK inhibitor PD0325901 are not totally convincing. I worry about nonspecific effects of this pharmacological reagent. Another inhibitor of cell proliferation like a Cochlea from a KO mouse would provide a second piece of evidence. How did the FGF inhibitor SU5402 effect curvature? In Figure 4C, I was not sure why do the measurements stop at 0.8 mm? It seems like the trend was going down which would make it not significant?

In the Discussion the authors state they "provided the first experimental evidence that nuclei stall at the luminal side of the pseudostratified epithelium during IKNM in normal development". I go back to the discrepancy between the simulation and experimental results in Figure 2J-M. They address this in the Materials and methods but some mention of it here would be appropriate.*Reviewer #3:*

The authors address many of the points raised in my review and the other reviewers. The section on the ERK waves and the mechanochemical feedback has improved considerably and is actually very nice. In particular the new data on the effect of blebbistatin is very nice and supports the model.

I do, however, have some problem with the first part of the manuscript on the interkinetic nuclear movement (IKNM). The main problem here is that the model they suggest ignore an essential aspect of the system which is that only some small fraction of the cells perform the luminal stalling at a given time. The model suggests that the cause for bending is that luminal stalling leads to an inverted wedge like morphology of the cells and thus leads to bending. However, in the model, luminal stalling is assumed to happen in all the cells. This does not seem to match the images in Figure 2A-C showing that most nuclei are closer to basal position. Hence, to model that the authors should have assumed that there are only few cells whose nuclei are stalled at the laminal position. It seems to me, that it is unlikely that such a model would produce significant bending in that case. In their response, the authors argue that their model like all mathematical models is a simplification of the real system. While it is true that models always simplify, it seems to me that the assumption that all cells stall is an essential divergence from the real system and cannot be simplified.

As I suggested previously, the nuclear stalling may actually be a result of bending and not the cause. I understand that this is hard to test experimentally. However, I feel that the current model maybe misleading.

Since I like the second part and find it interesting and more convincing, it may be worth for the authors to restructure their manuscript so that the ERK waves are at the beginning and are the main focus. The IKNM is an interesting observation that can be added with the two potential interpretations.

[Editors' note: further revisions were suggested prior to acceptance, as described below.]

Thank you for submitting your article "Retrograde ERK activation waves drive base-to-apex multicellular flow in murine cochlear duct morphogenesis" for consideration by *eLife*. Your article has been reviewed by Marianne Bronner as the Senior Editor, a Reviewing Editor, and three reviewers. No reviewers found for this submission.

The spiral shape of the mammalian cochlear duct is tightly coupled to its function in sound detection. However, the morphogenetic process that gives rise to the spiral structure is poorly understood. In this paper, by live-imaging of the developing cochlea from a FRET-based reporter mouse strain, Hiroshima et al. demonstrated elegantly that one of the mechanisms in cochlear duct elongation is driven by an oscillatory wave of ERK activity originating from the roof of the cochlear apex towards the base and a concomitant flow of cells from the base of the cochlear floor towards the roof of the apex.

All three reviewers felt the revised manuscript is much improved and concise by removing nuclear stalling data. The only suggestion is for you to consider a 3D schematic summary that illustrates direction of ERK oscillation, cell movement flow and the hot spot of cell proliferation in the floor of the cochlea. This will help readers outside of the ear field to appreciate your beautiful work.

Second, the citation of Bok et al., for reduction of cell proliferation at the base of the cochlear duct is incorrect. It is ok to state that cochlear duct growth requires Shh signaling but it is incorrect to say there is significant decrease in cell proliferation at the base of the cochlea in the Shh conditional mutants. First, only EdU-labeled hair cells were quantified in that paper, which could hardly account for the shortened cochlear duct phenotype. More importantly, those experiments were not designed to look at proliferation but cell cycle exit. They were conducted by injecting Edu at E13.5 and E14.5 and EdU-labeled hair cells were analyzed at birth. Heavily labeled Edu-positive cells at birth were cells that exited cell cycle shortly after EdU administration. Therefore, a reduced EdU labeling could mean there were more cells dividing at the base at E13.5 and E14.5 rather than less proliferation and the label were diluted and no longer detected by birth.

---

## [Author Response]

All three reviewers were impressed by the imaging results. Although the reviewers and editors find the concept and approach interesting, blocking cell proliferation may be too crude a method to address the authors' hypothesis and many questions were raised by the results of blocking cell division.

We understand that blocking cell proliferation is a rough experiment to address our hypothesis and admit its weakness as a methodological aspect in our study. We pondered over other alternatives among our current experimental techniques, but unfortunately, this particular method was the only possible approach. In response to the comments, we mitigated the claim in the section on nuclear behaviors and mentioned this limitation in the Discussion section.

Second, the relationship between cell proliferation and ERK-driven migration is also unclear. Please see comments from reviewer #2 and #3 for specifics.

We apologize for the lack of clarity on this. In response to the comments, we added the corresponding paragraph in the Discussion section.

Third, what is the relationship between SHH-induced proliferation and ERK activation as suggested by the authors (see comments from reviewer #1)?

In response to the comment by reviewer #1, we added the new results in Figure 5 as requested. Thank you for this suggestion. We believe that this point is made clear in the present manuscript.

Additionally, it is problematic to illustrate a difference in the bending force between medial and lateral cochlear duct that is presumably occurring at E12.5 and E14.5 with a cochlear dissection at E17.5. The tissue architecture is completely different between E12.5 and E17.5. The surgery basically removed a specific region of the cochlear duct, the stria vascularis, rather than medial versus lateral halves of the cochlear duct.

Thank you very much for your constructive comments. As the tissue structure is different between the E12.5 and E17.5, we also agree that it might be difficult to directly connect the surgery results obtained from the E17.5 cochlea and the force balance of the developing cochlea at earlier stages. Here, we would like to emphasize that we do not intend to link the ablation experiments and conjecture of mechanical forces between different stages. Rather, in this beginning section, we aim to introduce a motivation for examining the differences in the tissue architecture along the mediolateral axis in detail. We apologize for the confusion, and our original submission was unclear regarding this point. In the present manuscript, we added the following sentences: “This result motivated us to explore the differences in physical properties between the medial and the lateral side in earlier stages.”

Reviewer #1:This is a fascinating manuscript that explores for the first time the potential mechanisms underlying cochlear morphogenesis. The authors have used a combination of modeling, beautiful imaging and ERK-FRET reporter mice reporter mice to suggest at least two processes may be at play in cochlear shaping – differential interkinetic nuclear migration and a cellular flow that appears to correlate with ERK activation.I have no major concerns with this lovely piece of work. The imaging and quantification is meticulous, and the observations made by the authors are novel and will of great interest to cell biologists interested in morphogenesis, no just aficionados of the inner ear.The one suggestion I would make is for the authors to clarify the relationship between cell proliferation and ERK activation. When they reference the inner ear literature, they point out that FGF pathway mutants have deficient cochlear morphogenesis and proliferation, and they hypothesize that FGF-induced ERK activation may be responsible for their propagating waves. However, they also reference work suggesting that cellular extension during collective migration can also induce ERK activation and also suggest SHH-induced proliferation as another causative factor in promoting ERK activation through proliferation. I think the authors should try and clarify this – both in their explanation, but also by comparing the effects of the MEK inhibitor PD0325901 on ERK activity and tissue flow speed (Figure 4I and Figure 3—figure supplement 1F) with the effects of the FGFR inhibitor SU5402, and also Shh inhibitors like cyclopamine. If the effects they see are directly due to FGF signaling, one would expect a change in ERK activation and cell flow with the same kinetics as with PD0325901. However, if Shh-induced proliferation is responsible, the change in ERK activation would take much longer to achieve. I think these experiments should be possible to do in a relatively short period of time.

We appreciate you raising an important point. As you mentioned, we have argued that ERK activation requires cell migration flow and that SHH-induced proliferation is another underlying causative factor in promoting the ERK-migration feedback system. We agree that this point was unclear in our original submission.

In accordance with your comment, we performed an inhibitor assay with the FGFR inhibitor SU5402 and the Shh signaling inhibitor cyclopamine. Treatment with SU5402 resulted in a significant decrease in tissue flow speed along with ERK inactivation immediately after administration (3 hours), and the level was almost the same as that of treatment with the MEK inhibitor PD0325901. This indicates that ERK-driven cell migration is directly due to FGF signaling. However, for the cyclopamine treatment, we found that the degree of decrease in cell flow was not as significant as that of the MEK inhibitor treatment within 3 hours after administration and also observed no change in ERK activation. It took 8 hours to reach the same degree of speed decrease as the MEK inhibitor treatment. These results support that SHHinduced proliferation was less influential on ERK activation in a short time period but was prominent through cell proliferation over longer periods. We have added the corresponding new results in Figure 5 and the comment in the Discussion in response to your comments.

Reviewer #2:The paper by Hirashima and colleagues shows some interesting cellular mechanisms they conclude drive the spiraling and outgrowth of the mammalian cochlea. The two cellular mechanisms they propose are supported by experiments and modeling. The spiraling ERK wave and the contrasting movement of lateral cells was very intriguing. However, the ERK wave and lateral cell movements seem disconnected from the bending forces discussed. Are the authors saying that the ERK mediated lateral cell movements are important for cochlear growth while the MEL is important for the bending? The two mechanisms they discuss seem insufficient to explain all of cochlear spiraling. Other cellular mechanisms such as cell proliferation and convergent extension are mentioned but their roles are not incorporated into their Discussion. Are they not required? How do they complement their results?

We thank the reviewer for bringing up many important questions, and we agree that our proposal might not be clear enough in the original submission. We read several questions and concerns that were rephrased in the following detailed comments. Please see our reply below. We believe that our present manuscript has been made more convincing and hope it meets your criteria.

1) While the authors talk about bending forces, the paper has no measurements of the forces generated by different tissues. I also feel there are other cellular mechanisms that are mentioned but never incorporated into their proposed explanation for duct coiling such as convergent extension and actomyosin based basal shrinkage. Proliferation is discussed quite a bit but seems to be dismissed as a force. In the introduction they mention how Shh mediated proliferation is required for duct elongation while Fgf10 null mutants have a shortened duct yet normal proliferation. So what is the role for proliferation? Maybe they can answer this in the context of their interesting observation that there is more proliferation in the roof than the floor which would be predicted to bend the cochlea along that axis. When combined with the medial lateral bending could these two forces result in the spiraling? It also seems like this differential proliferation between the floor and roof was in more than just the epithelium, correct? Could the cartilaginous capsule around the duct guide the bending as well? In their culture experiments, if too much of the capsule was removed then normal duct development was disrupted.

As you point out, the previous manuscript included fewer arguments of active force generation despite the fact that it should be a key link between cellular processes and tissue morphogenesis. In the revised manuscript, we improved this point by supplementing new data on cellular force generation.

Regarding convergent extension, we manually tracked each cell movement in 3D and examined the trace lines of neighboring cell clusters. As a result, we were not able to find any evidence that mediolateral convergent extension occurs during stages E12.5 to E14.5. This result is in accordance with previous studies demonstrating the existence of convergent extension only after E14.5 as described in the Introduction. We added this to Figure 5—figure supplement 3. However, we were unable to perform the complete automatic 3D single-cell tracking as described in the reply to your comment #4, meaning that we cannot completely deny the existence of convergent extension in our observation. We included this concern in the Discussion as a possible extension of this study.

Meanwhile, we have argued that actomyosin-based shrinkage plays a central role in cell migration. Thanks to your comments, we have added a new section and dataset, showing the active shrinkage during cell migration from the base to the apex in Figure 6. We believe that this provides further understanding of the mechano-chemical feedback system of ERK activation and cell flow during the development of murine cochlear ducts.

This reviewer gives a thought-provoking comment about spiral formation. We consider that cell proliferation underlies global tissue growth through volumetric local growth but is not the sole factor that produces mechanical forces underlying the asymmetry along both the mediolateral axis and the roof-floor axis. Rather, we have argued that the “hotspot” where the remarkable cell proliferation occurs in the basal region plays a role as a source of material supply to the growing apex side of the cochlear duct as mentioned in the text.

We consider that the cartilaginous capsule may play an essential role in cochlear morphogenesis, especially as a physical restriction to avoid the duct growing outward. However, in this case, a detailed balance of overall tissue growth between the epithelium and the capsule is essential, and elucidation of this mechanism is beyond the scope of this paper. In addition, we do not have data that strongly support that the capsule controls the spiral formation. Therefore, we refrain from pursuing further on this point, and included these comments in the Discussion section to provide our idea to the readers.

2) Their demonstration that the bending forces are in the medial half is interesting but the only tissue whose mechanism is studied is the MEL. Could convergent extension in other medial tissues such as the prosensory domain (which Wang et al., showed was occurring in this tissue) and surrounding mesenchyme be the main force generator for the bending of the medial half of the cochlear duct? Does the MEL cultured by itself bend? They say that cell intercalation can drive ductal elongation but not bending (Introduction) but can't convergent extension occur asymmetrically in the tissue? Such as by occurring in the overlying medial mesenchyme but not in the medial epithelium. It should be noted that the bending by the epithelium does not have to provide high forces as long as the force provided by other tissues are similar across the medial lateral axis, the bending in the epithelium could bias the mass of tissue to bend.

We attempted to isolate the medial half of the cochlear duct from E12.5 to E14.5 but failed to properly dissect the tissues via manual surgery. We also tried to determine the dynamics of surrounding mesenchymal cells, but there appeared to be an insufficient number of mesenchymal cells to be perceived as group cell behaviors. As far as our observations, we could not see the asymmetric cell intercalation as reported in the development of *Drosophila*’s genital disc (Sato et al., 2015). Again, we have no positive evidence to support the existence of convergent extension in the cochlear epithelium as replied to your comment #1. Additionally, there are no reports on asymmetric convergent extension of the overlying medial mesenchyme as far as we know. As discussed above, we only mentioned this issue in the Discussion section.

3) The mathematical modeling for the luminal bending is less convincing than the mathematical modeling for the ERK and Cell flow coupling. The simulated curves in Figure 2K are quite different from the Experimental measure in Figure 2M, especially for the Mitomycin C condition. I feel that the values plugged in for the Numerical simulation, the standard parameter set were not well justified. What happened to the simulations as these values changed? Was the parameter space for acceptable values broad? In contrast the parameters for the numerical simulation of the ERK activation waves and cell flows were well justified. The parameters chosen might explain the big differences between simulation and experimental in Figure 2.

Thank you for your constructive suggestions. Although we have already chosen the standard set of parameter values under which the simulated results satisfactorily mimic the experimental observations, we did not show the degree to which the values were justified as you point out. According to your suggestions, we demonstrated the similarity of the simulated results to the experimental observation by introducing the root-mean-square error (RMSE), which measures the differences in datasets between simulations and experiments. As a result, the RMSE at the standard parameter set showed a minimal value and small variance, which indicates that the parameter values we used are plausible for the conditions of numerical simulation. These results are shown in Figure 2—figure supplement 1B.

We have also been concerned with the quantitative difference in the curves between experiments and simulations for the Mitomycin C condition. Because all parameters are determined either by measurement or fitting for the control condition, we consider that this difference largely stems from the boundary conditions of the model. In the simulations, the edges of cells at the tissue boundary were set to be free in the finite window size, which was determined as a practical condition of the numerical investigation. On the other hand, in a real experimental situation, the boundary cells are constrained by the other cells outside the corresponding window. We admit that this point is important to the reader. We have included these statements in the Materials and methods section.

4) For the cell tracking experiments in the lateral region the resolution was 4-5 cells. The resulting cell flow patterns were very interesting but why didn't the authors track single cells? Segmenting individual cells via cytoplasmic labeling is much trickier but the nuclei are identifiable and the Imaris software they used in the paper has a cell tracking feature for such labeling. I would think that individual cell movements might provide more insights. In subsection “Retrograde helical ERK activation waves drive base-to-apex multicellular flow” they say they can see cell contractions which I assume is for individual cells? How were cell contractions identified? Video 5 was excellent and very informative. Do the cell flows correlate at all with the proliferation seen with Edu staining?

We worked on single-cell tracking from nuclei reporter images (ubiquitous H2B-mCherry expression) but found that it failed to acquire reliable tracking data mainly due to weak signals and densely packed nuclei, even using Imaris software. As you suggest, we realize that singlecell tracking should provide more insights despite the challenge, and included this comment in the Discussion section.

In accordance with the comment regarding cell contraction, we added a new section and dataset for an inhibitor treatment of myosin activity and local tissue deformation owing to cell contraction as shown in Figure 6. We believe that this supplementation strengthens the validity of our claim. We appreciate your constructive suggestion.

As for the correlation between the cell flows and proliferation, we agree that it will provide further profound information to the readers. The corresponding movie is indeed informative, but it is almost impossible to recognize cell proliferation. In addition, because it is not easy to link static EdU images with movies, we refrain from adding results in the text.

Reviewer #3:1) The authors argue that cell cycle arrest results in a decrease in the curvature of the cochlear duct, which supports the hypothesis that luminal nuclear stalling promotes MEL bending. This is fine, but luminal nuclear stalling can be a result and not a cause. Since in a bent region, the basal side is more packed, this density gradient can be the cause of nuclei stalling at the luminal side. The fact that the curvature decreased but not diminished after cell cycle arrest could suggest that nuclear stalling is not required for bending, but rather reinforces it.

We like this idea, and in fact, we mentioned the possibility that luminal nuclear stalling can be a result in the previous manuscript: “…while it remains unclear whether luminal nuclear stalling results from the MEL curvature, that is, the convexity of the luminal side”. It might be difficult to completely diminish the curvature because the epithelial tissue has hysteresis to some extent, even with cell cycle arrest. Luminal nuclear stalling promotes MEL bending, and the nuclear density gradient along the luminal-basal axis can cause luminal nuclei stalling. Therefore, we consider that there would be feedback in the nuclear movement and tissue geometry. We agree with your idea, and thus supplemented some sentences in the Discussion section.

2) Since the authors discuss both cell proliferation and nuclear stalling, and cell migration, as forces that can drive bending and coiling, it hard to interpret the results of the mitomycin C experiment. Could it be that the tissue is less curved because there are less cells to supply the elongation tissue rather than less nuclear stalling? The authors should consider inhibiting either cell migration or the cytoskeletal machinery required for IKNM to dissect these effects.

We admit that blocking cell proliferation is a rough experiment and is weak to distinguish which cellular processes are critical to morphogenetic mode at the tissue scale. We carefully considered other alternatives among our current experimental techniques, but unfortunately, we had no elegant method to suppress nuclear movement in the MEL, cell proliferation in the specific region, or cell migration from the base to the apex. We apologize for not being able to experimentally respond to your comment. Please understand that this is a very challenging experiment in our current situation. We have softened the claim in the section on nuclear behaviors and added a new paragraph concerning the methodological limitations of this study in the Discussion section.

3) The authors present a mathematical model to demonstrate that nuclear stalling in the luminal side results in bending. To model nuclear motion they use a parameter, γ, which controls the degree of basalward movement after IKNM. Modeling in such way means that other than γ=1, the nucleus never fully returns to the basal side, but if I understand correctly this is not the case, as even if the nuclei that stall at the luminal side, eventually return to the basal side.

Thank you for your careful reading. We set γ=0.9 as the standard parameter value in the numerical simulation by considering the size of the nucleus and based on observation. This means that the nuclei fully return to the basal side according to the cell cycle length in the simulation, which occurs in the experiment only in the uncurved region of the MEL. However, as shown in Figure 2D’, we never observe nuclear return to the edge of the basal side in the curved region within our observation period. To clarify this point, we added the above sentence in the Materials and method section.

4) Furthermore, for luminal nuclear stalling, the authors tracked only the nuclei of dividing cells. This makes the data in Figure 2D' much clearer. However, in their model the authors show only these nuclei and not all nuclei. In addition, they show many crowded nuclei in the model, yet this is not observed in the images provided in the manuscript. Therefore, it seems the model does not represent the morphology of the tissue properly. The authors should model the process with non-dividing cells at the basal side.

This comment is important for considering the use of the mathematical model. It should be emphasized that we do not aim to incorporate the physical situation precisely but rather utilized it to demonstrate the concept in some ideal situations. In any case, mathematical modeling is an abstraction of reality, and we hope you will agree with this. Although our model does not include detailed tissue architectures and physical properties, it does include some experimentally acquired parameters, such as the dimensions of the cell. Hence, we are convinced that the model captures the essential aspects of this complex process and is useful for delivering the concept.

Another possible approach to express individual nuclear dynamics is a particle-based model, which ignores cell shape and focuses only on the center of mass of the nuclei. For this model, it is not easy to link cell dynamics and mechanical integrity of the epithelial tissues with densely packed nuclei–this is a challenge for future studies. We have included these comments in the Materials and methods section. We believe that this response to your comment will be helpful to our readers, especially those interested in the use of mathematical models.

5) In subsection “Spatial heterogeneity of cell proliferation suggests cellular inflow to the lateral side of the apex to realize cochlear bending” the authors claim that the higher volumetric growth measured at the MEL should cause an opposite curvature relative to the innate one. This is true if EdU intensity is proportional to volumetric growth, but cells in the MEL and LEL may not be the same size. For example, cells in the MEL could be smaller than cells in the LEL. The authors should therefore measure the nuclei number density and the volumetric cell density to clarify this. If the number density of the nuclei is indeed higher at the MEL, it may also explain the higher structural integrity of the MEL relative to the LEL demonstrated in Figure 1C.

This is an important issue when considering the relationship between local volumetric growth and tissue morphogenesis. In accordance with this comment, we measured the nuclei density on both the floor and roof sides and found that the nuclei density was not significantly different along the mediolateral axis. We included the data in Figure 3. This observation supports the idea that the EdU intensity distribution is worthy of consideration as a marker of local tissue growth at a fixed moment in time due to cell proliferation. However, EdU intensity measurement is not the sole way to evaluate local tissue growth over time. We included this concern in the Materials and methods section. We would also like to emphasize that the conclusion of the EdU experiments section is not a positive statement to support the main driver of cochlear morphogenesis; rather, we regard this section as a bridge passage.

6) The authors show the effect of ERK inhibition on tissue flow speed. This is a very important observation and raises several important questions. What is effect of ERK inhibition on curvature? On tissue length? On proliferation? These will provide a more complete understanding of the effect of RK inhibition.

Since we have agreed that these comments are critical to understanding the ERK activity of cochlear morphogenesis, we added a new section with a dataset regarding the effect of ERK inhibition on the multiple aspects in Figure 4 of the present manuscript. We believe that these findings provide fundamental information on the long-term effects of ERK inactivation on cochlear duct morphogenesis.

7) The authors should also test the effect of mitomycin C on cell flow and ERK activity. As mentioned above, it is not clear whether the effect of mitomycin C is a result of less nuclear stalling or perhaps less cells that flow towards the apex.

We carefully considered whether our current experimental techniques are truly able to resolve your concerns. Mitomycin C treatment certainly results in both less nuclear stalling and less cell flow towards the apex, as we argue that cell flow is driven by the heterogeneous cell proliferation profile. Unfortunately, this experiment is the best possible option for us at the present time. We have accepted the weakness of the mitomycin C experiments, and this concern was included in the Discussion section.

8) In Figure 3 the authors analyze the EdU distribution over the cochlear duct. This analysis is done using the maximum intensity projection of the stack. It seems that a more accurate way to quantify would be to use the summed intensity image rather than the maximum intensity image. This may reveal additional details that were missed by throwing away all other layers except the one at maximum intensity.

In response to this comment, we made three different versions of the projection data: (1) maximum intensity projection, (2) summed intensity projection, and (3) mean intensity projection averaged within the cochlear duct. All processed data for the three experiments were added to Figure 3—figure supplement 1. As a result, we found no qualitative differences among different projections. Since we are not sure whether the summed intensity projection is better than the maximum intensity projection and the readers can refer to the different versions in the supplementary figure, we maintained the previous version in the current manuscript aside from the mapped graphs, which were replaced with the sample mean data over three samples instead of the single data.

9) In Figure 4 the colors used for the ERK activity analysis are very hard to see for color-blind people. It would be easier for this audience if the authors changed one of these colors to green/red/yellow.

This is an important comment for a wide range of readers. We believe that the current coloring is the best for effectively illustrating the spatio-temporal patterns of ERK activity to the readers. However, we admit that it is difficult to see some figures, especially the change of color in Figure. Therefore, we included another version for color-blind people below the corresponding figures.

[Editors' note: further revisions were suggested prior to acceptance, as described below.]

All three reviewers have judged that your manuscript is of interest and represents an advancement to the field. However, the manuscript cannot be accepted at its current form because of the issue whether IKNM described is causal or consequential to cochlear bending. This issue needs to be better resolved. Either the mathematical model is revised to accommodate for fewer cells stalling at the luminal side but could still account for the cochlear bending or more cells can be observed to stall at the luminal side that could account for the cochlear bending. Alternatively, the entire manuscript can be totally revised to accommodate for these two possibilities in an unbiased manner, starting with the Title.

We would like to thank the reviewers for their thoughtful responses and efforts toward improving our manuscript. We have responded to all concerns and revised the manuscript according to the suggestions. As described in the cover letter, we agree with the reviewers’ suggestions and totally changed the whole structure of our manuscript. In the revised version, the part of apical nuclear stalling was removed and the ERK activation wave becomes at the center. With this, we changed the Title as “Retrograde ERK activation waves drive base-to-apex multicellular flow in murine cochlear duct morphogenesis”. We believe that this change makes the main point of the paper clearer, and the revised manuscript meets with your approval.

Since many researchers have temporarily lost access to the labs, we will give authors as much time as they need to submit revised manuscripts. We are also offering, if you choose, to post the manuscript to bioRxiv (if it is not already there) along with this decision letter and a formal designation that the manuscript is "in revision at eLife". Please let us know if you would like to pursue this option. (If your work is more suitable for medRxiv, you will need to post the preprint yourself, as the mechanisms for us to do so are still in development.)Specific comments from two of the reviewers are listed below as guidelines for your revision:Reviewer #2:The authors have addressed most of my comments. The experiments with the MEK inhibitor PD0325901 are not totally convincing. I worry about nonspecific effects of this pharmacological reagent. Another inhibitor of cell proliferation like a Cochlea from a KO mouse would provide a second piece of evidence. How did the FGF inhibitor SU5402 effect curvature? In Figure 4C, I was not sure why do the measurements stop at 0.8 mm? It seems like the trend was going down which would make it not significant?

We appreciate your suggestion. We performed pharmacological assays using the SU5402 and found that the cochlear duct length became shorter and the curvature profile became overall larger compared with the control as observed in the PD0325901 treatments (Figure 2A-D). This reviewer pointed out the difference of measurement range between the control group and the inhibitor treatment group. This comes from the fact that the cochlear duct length was significantly shortened by the inhibitor treatments. In the current version, we adopted the curvature profile along the relative arc length normalized by the total arc length as the main figure (Figure 2C) because the size of cochlear duct became each treatment. We believe that this graph representation would be easier for readers to evaluate. We included these statements in the Introduction and Figure 2.

In the Discussion the authors state they "provided the first experimental evidence that nuclei stall at the luminal side of the pseudostratified epithelium during IKNM in normal development". I go back to the discrepancy between the simulation and experimental results in Figure 2J-M. They address this in the Materials and methods but some mention of it here would be appropriate.

In response to the previous reviewer comments, we deleted the whole part of apical nuclear stalling because we admit that the experimental evidences on apical nuclear stalling in the medial side of cochlear duct are rather weak to support it as a main driving factor for the duct bending. With this, this comment is not applicable anymore. We thank for your efforts on this comment.

Reviewer #3:The authors address many of the points raised in my review and the other reviewers. The section on the ERK waves and the mechanochemical feedback has improved considerably and is actually very nice. In particular the new data on the effect of blebbistatin is very nice and supports the model.I do, however, have some problem with the first part of the manuscript on the interkinetic nuclear movement (IKNM). The main problem here is that the model they suggest ignore an essential aspect of the system which is that only some small fraction of the cells perform the luminal stalling at a given time. The model suggests that the cause for bending is that luminal stalling leads to an inverted wedge like morphology of the cells and thus leads to bending. However, in the model, luminal stalling is assumed to happen in all the cells. This does not seem to match the images in Figure A-C showing that most nuclei are closer to basal position. Hence, to model that the authors should have assumed that there are only few cells whose nuclei are stalled at the laminal position. It seems to me, that it is unlikely that such a model would produce significant bending in that case. In their response, the authors argue that their model like all mathematical models is a simplification of the real system. While it is true that models always simplify, it seems to me that the assumption that all cells stall is an essential divergence from the real system and cannot be simplified.As I suggested previously, the nuclear stalling may actually be a result of bending and not the cause. I understand that this is hard to test experimentally. However, I feel that the current model maybe misleading.Since I like the second part and find it interesting and more convincing, it may be worth for the authors to restructure their manuscript so that the ERK waves are at the beginning and are the main focus. The IKNM is an interesting observation that can be added with the two potential interpretations.

In response to your comments, we determined to change the whole structure of the manuscript. We admit that the experimental evidences on apical nuclear stalling in the medial side of cochlear duct are rather weak to support it as a main driving factor for the duct bending. We also agreed your idea that ERK waves should be the main focus in the revised manuscript. We believe the updated manuscript is more convincing and meets your criteria.

[Editors' note: further revisions were suggested prior to acceptance, as described below.]

The spiral shape of the mammalian cochlear duct is tightly coupled to its function in sound detection. However, the morphogenetic process that gives rise to the spiral structure is poorly understood. In this paper, by live-imaging of the developing cochlea from a FRET-based reporter mouse strain, Hiroshima et al. demonstrated elegantly that one of the mechanisms in cochlear duct elongation is driven by an oscillatory wave of ERK activity originating from the roof of the cochlear apex towards the base and a concomitant flow of cells from the base of the cochlear floor towards the roof of the apex.All three reviewers felt the revised manuscript is much improved and concise by removing nuclear stalling data. The only suggestion is for you to consider a 3D schematic summary that illustrates direction of ERK oscillation, cell movement flow and the hot spot of cell proliferation in the floor of the cochlea. This will help readers outside of the ear field to appreciate your beautiful work.

Thank you for your suggestion. We modified Figure 3L, which will be helpful for readers.

Second, the citation of Bok et al., for reduction of cell proliferation at the base of the cochlear duct is incorrect. It is ok to state that cochlear duct growth requires Shh signaling but it is incorrect to say there is significant decrease in cell proliferation at the base of the cochlea in the Shh conditional mutants. First, only EdU-labeled hair cells were quantified in that paper, which could hardly account for the shortened cochlear duct phenotype. More importantly, those experiments were not designed to look at proliferation but cell cycle exit. They were conducted by injecting Edu at E13.5 and E14.5 and EdU-labeled hair cells were analyzed at birth. Heavily labeled Edu-positive cells at birth were cells that exited cell cycle shortly after EdU administration. Therefore, a reduced EdU labeling could mean there were more cells dividing at the base at E13.5 and E14.5 rather than less proliferation and the label were diluted and no longer detected by birth.

We revised as follows according to Discussion of the referenced paper:

“Deletion of Shh expression leads to a shortening of the cochlear duct, and it was proposed that the SHH signaling promotes growth of the cochlear duct mainly in the base region (Bok et al., 2013).”.